

# Potential and limits of vegetation indices compared to evaporite mineral indices for soil salinity discrimination and mapping

Abderrazak Bannari [1] and Abdelgader Abuelgasim [2]

[1] Space Pix-Map International Inc., Gatineau (Québec) J8R-3R7, Canada. Email: abannari@bell.net

[2] Department of Geography and Urban Sustainability, United Arab Emirates University, AL-Ain 15551, United Arab Emirates. Email: a.abuelgasim@uaeu.ac.ae

*Correspondence to*: abannari@bell.net

**Abstract.** The study aims to analyze the ability of the most popular and widely used vegetation indices (VI's), including NDVI, SAVI, EVI and TDVI, to discriminate and map soil salt contents compared to the potential of evaporite mineral indices such as SSSI and NDGI. The proposed methodology leverages on two complementary parts exploiting simulated and imagery data acquired over two study areas, i.e. Kuwait-State and Omongwa salt-pan in Namibia. In the first part, a field survey was conducted on the Kuwait site and 100 soil samples with various salinity levels and contents were collected; as well as, herbaceous vegetation cover canopy (alfalfa and forage plants) with various LAI coverage rates. In a Goniometric-Laboratory, the spectral signatures of all samples were measured and transformed using the continuum removed reflectance spectrum (CRRS) approach. Subsequently, they were resampled and convolved in the solar-reflective spectral bands of Landsat-OLI, and converted to the considered indices. Meanwhile, soil laboratory analyses were accomplished to measure pHs, electrical conductivity (EC-$_{Lab}$), the major soluble cations and anions; thereby the sodium adsorption ratio was calculated. These elements support the investigation of the relationship between the spectral signature of each soil sample and its salt content. Furthermore, on the Omongwa salt-pan site, a Landsat-OLI image was acquired, pre-processed and converted to the investigated indices. Mineralogical ground-truth information collected during previous field work and an accurate Lidar DEM were used for the characterization and validation procedures on this second site. The obtained results demonstrated that regardless of the data source (simulation or image), the study site and the applied analysis methods, it is impossible for VI's to discriminate or to predict soil salinity. In fact, the spectral analysis revealed strong confusion between signals resulting from salt-crust and soil optical properties in the VNIR wavebands. The CRRS transformation highlighted the complete absence of salt absorption features in the blue, red and NIR wavelengths. As well as the analysis in 2D spectral-space pointed-out how VI's compress and completely remove the signal fraction emitted by the soil background. Moreover, statistical regressions ($p < 0.05$) between VI's and EC-$_{Lab}$ showed insignificant fits for SAVI, EVI and TDVI ($R^2 \leq 0.06$), and for NDVI ($R^2$ of 0.35). Although the Omongwa is a natural flat salt playa, the four derived VI's from OLI image are completely unable to detect the slightest grain of salt in the soil. Contrariwise, analyses of spectral signatures and CRRS highlighted the potential of the SWIR spectral domain to distinguish salt content in soil regardless of its optical properties. Likewise, according to Kuwait spectral data and EC-$_{Lab}$ analysis, NDGI and SSSI incorporating SWIR wavebands have performed very well and similarly ($R^2$ of 0.72) for the differentiation of salt-affected soil classes. These statistical results were also corroborated visually by the maps derived from these evaporite indices over the salt-pan site, as well as by their



consistency with the validation points representing the ground truth. However, although both the indices have
mapped the salinity patterns almost similarly, NDGI further highlights the gypsum content. While the SSSI show
greater sensitivity to salt crusts present in the pan area that are formed from different mineral sources (i.e., halite,
gypsum, etc.).

## 1. Introduction

Soil salinity or salinization is a global environmental threat, it occurs in different geographical zones characterized
by different climatic conditions and can result from both natural and anthropogenic actions (Shahid et al., 2018). In
humid zones, rainfalls exceeds the evaporation, thus the soluble salts are leached from the soil surface to deeper
zone. While, semi-arid and arid lowlands are more affected because of near surface saline groundwater and due to
evaporation exceeding precipitations (Dehaan and Taylor, 2002; Shahid and Rahman, 2011). Moreover, soil salinity
is associated with several other physical factors including soil properties, permeability, geomorphology, geology,
micro-topography, wastewater use and climate variability (Hartemink, 2014; Shahid and Behnassi, 2014; Dagar et
al., 2016; Bannari and Al-Ali, 2020; Bannari et al., 2021). During the past decades the global warming has
decreased precipitations, increased temperatures, reduced soil moisture regime and, subsequently, accelerated
expansion of this menacing phenomenon. Indeed, it represents a serious problem for health and functionality of arid
ecosystems, significant impacts on land desertification, reduction of crop production and economic aspects
(Mougenot et al., 1993; Naing'OO et al., 2013; Arrouays et al., 2017; FAO, 2018; Ivushkin et al., 2019; Hassani et
al., 2020); as well as on human wellbeing and sustainable development. Whereas, in irrigated agricultural lands,
salinity occurs when salts are concentrated in soils by the evaporation of irrigation water. The major causes are a
combination of poor land management and crude irrigation practices, which cause changes in soil and vegetation
cover, and ultimately loss of vegetation and agricultural productivity (Metternich and Zinck, 2003; Masoud and
Koike, 2006; Corwin and Scudiero, 2019; Zhu et al., 2021; Gopalakrishnan and Kumar, 2021). Obviously,
combating soil salinization should lead to enhance soil fertility, agricultural productivity and profitability, and
ensure food security (Teh and Koh, 2016).

Furthermore, it is common that both saline and sodic conditions occur together in the soil. Salinity refers to the

amount of soluble salts in soil, such as sulfates ($SO_4$), carbonates ($CO_3$), and chlorides ($Cl^-$) mainly of sodium (Na),
Calcium (Ca), Potassium (K), Magnesium (Mg) and other cations to a lesser extent (Richards, 1954). The solubility
of halite (NaCl), calcium sulphate-anhydrite ($CaSO_4$) and gypsum ($CaSO_4.2H_2O$) is used as a standard for
comparing the levels of salinity content in the soil. According to Richards (1954), a soil is said to be saline when it
has an electrical conductivity of saturation extract (ECe) greater than 4 dS.m$^{-1}$ at 25°C and a pHs < 8.2. While
sodicity refers to the exchangeable sodium ($Na^+$) relative to exchangeable $Ca^{2+}$ and $Mg^{2+}$ in soil. Sodicity has a
strong influence on the soil structure, dispersion occurs when the clay particles swell strongly and separate from
each other on wetting. On drying, the soil becomes dense, cloddy, and without structure (Charters, 1993; Sumner et
al. 1998). Sodic soils have a pHs greater than 8.2 and a preponderance of sodium, carbonate and bicarbonate
(Richards 1954). Ranges of salinity are usually described as non-saline, very slightly saline, slightly saline,
moderately saline, and strongly saline (high to extreme) based on the ECe values (USDA, 2014; Metternicht and



Zinck, 1997; Soil Science Division Staff, 2017). Traditionally, soil salinity is measured by geophysical method
(EM38) in the field (apparent salinity) and through laboratory determination (EC-$_{Lab}$) using water extracted from a
saturated soil paste which is globally accepted a standard to quantify soil salinity (Norman, 1989; USDA, 2004 and
2014; Zhang et al., 2005). Unfortunately, the laboratory method is expensive, time-consuming, and laborious when
large area is to be investigated, especially for temporal salinity monitoring. Thus, remote sensing science,
technology and image processing methods have outperformed ground-based methods, and they have been used for
identifying, mapping and monitoring salt-affected zones (Masoud and Koike, 2006; Meternich and Zinck, 2009;
Ben-Dor et al., 2009; Nawar et al., 2014; Wu et al., 2014; El-Battay et al., 2017; Bannari et al., 2018 and 2020;
Bannari, 2019; Davis et al., 2019; Al-Ali et al., 2021).
Previously, photo-interpretation approaches have been adopted to follow the development and the dynamics of
soil salinity and sodicity in space and time (Manchanda and Khanna, 1979; Rao et al., 1991). These approaches have
been based on the analysis of colour-infrared photographs or on the false color composites of images acquired from
space with the first generation of Landsat sensors (MSS and TM). Nevertheless, the advancements in multispectral,
hyperspectral, thermal, and radar technologies with significant radiometric performances and high signal-to-noise
ratio (SNR) are providing the best and the newest opportunities for more precise and more effective salinization
detection and prediction (Dehaan and Taylor, 2002; Metternich and Zinck, 2003; Lasne et al., 2009; Fan et al.,
2016; Nurmemet et al., 2018; Abuelgasim et al., 2018; Hoa et al., 2019; Wang et al., 2020). Indeed, thanks to the
free availability of remote sensing data acquired with different sensors onboard various platforms, soil salinity was
modeled for global, regional and local scales using, respectively, coarse, moderate and high spatial resolutions, i.e.,
MODIS, Landsat, Sentinel, Ikonos, and Worldview (Shamsi et al., 2013; Alexakis et al., 2016; Bannari et al., 2017a;
Kasim et al., 2018; Whitney et al., 2018; Ivushkin et al., 2019; Bannari, 2019; Moussa et al., 2020; Hassani et al.,
2020; Al-Ali et al., 2021). However, the most frequently used data to investigate and map soil salinity remain those
acquired by remote sensing sensors with medium spatial and spectral resolutions, such as Landsat series (TM,
ETM+, and OLI) and Sentinel-MSI (Joshi et al., 2002; Metternich and Zinck, 2009; Fan et al., 2016; El-Battay et
al., 2017; Bannari et al., 2018 and 2020; Davis et al., 2019; Taghadosi et al., 2019; Wang et al., 2019).
Otherwise, in addition to remote sensing sensors technologies improvement and innovation, numerous image
processing approaches and models were also developed and applied for soil salinity retrieval. They include mixture-
tuned matched filter approach (Dehaan and Taylor, 2003), regression of multi-spectral bands (Lobell et al., 2010;
Fan et al., 2012; Sidike et al., 2014), partial least square regression (Fan et al., 2015; Wang et al., 2018;
Gopalakrishnan and Kumar, 2020), multivariate adaptive regression splines (Nawar et al., 2015), artificial neural
network model (Farifteh et al., 2008; Jiang et al., 2019; Boudibi et al., 2021), linear spectral mixture analysis (Ghosh
et al., 2012; Masoud et al., 2019), spectral angle mapper (Bharti et al., 2015; Wang et al., 2021), support vector
machines (Gleeson et al., 2010; Jiang et al., 2019), and machine learning regression (Wu et al., 2018; Hassani et al.,
2020). Definitely, these sophisticated and complicated methods require extensive training information and/or ground
endmembers measurements. However, the simplicity of empirical and/or semi-empirical methods based on spectral
indices are easier to transfer between sensors and can be used as a robust alternative compared to the revolutionary
and complex modelling methods; because they are based on the knowledge of spectral absorption features that



characterize specifically the target under investigation (Rouse et al., 1974; Peon et al., 2017, Milewski et al., 2019).
Moreover, they have the advantage of being reproducible, easily transferable and applicable in other geographic
regions (Mulder et al., 2011).
In the literature, some evaporite mineral indices have been proposed for soil salinity detection and mapping. For
instance in Pakistan, Khan et al. (2001) proposed three soil salinity indices based on red and near-infrared (NIR)
bands of LISS-II sensor onboard the Indian satellite IRS-1B. These indices are named Brightness Index (BI),
Normalized Difference Salinity Index (NDSI) and Salinity Index (SI). Among them, the authors found that the
NDSI is the most promising for different salinity classes' characterization in semi-arid environment using satellite
images and *in situ* observations. In irrigated agricultural land in Syria, Al-Khaier (2003) highlighted the importance
of shortwave infrared (SWIR) bands of Landsat-ETM+ and ASTER for soil salinity contents discrimination. He
proposed the Salinity Index (SI-$_{ASTER-4,5}$) based on bands 4 and 5 of ASTER sensor (i.e., B4: 1.6-1.7 μm and
B5:2.145-2.185 μm) or the bands 5 (SWIR-1) and 7 (SWIR-2) of Landsat-ETM+. Based on the field soil sampling
and EC-$_{Lab}$, the validation of this index showed a very good potential for salt-affected soil prediction. Moreover, in
the context of a cooperative project between India and the Netherlands (IDNP, 2002) three soil salinity indices were
proposed. These indices integrate the NIR and SWIR bands of Landsat-TM, and are named SI-1, SI-2 and SI-3.
Combining field soil survey, soil chemical laboratory analysis, spectroscopy measurements and ALI-EO-1 image,
Bannari et al. (2008a and 2016) demonstrated that the SWIR bands are more sensitive than other bandwidths to
discriminate among different soil salinity classes, particularly slight and moderate salinity in irrigated agricultural
lands. Consequently, they proposed the Soil Salinity and Sodicity Index (SSSI) integrating the SWIR bands of ALI-
EO or Landsat-OLI sensors. Recently, based on the gypsum absorption feature in 1.75 μm and following the same
concept behind the development of normalized difference vegetation index (NDVI), Milewski et al. (2019) proposed
the normalized difference gypsum index (NDGI). This new index exploits the most relevant narrow wavelengths
characterizing the gypsum absorption features: 1690 and 1750 μm. It has been tested on Omongwa salt-pan area in
Namibia, which is a natural flat salt playa dominated by evaporite minerals such as halite, gypsum, calcium
carbonate, and minor content of clay (Mees, 1999; Fookes and Lee, 2018; Genderjahn et al., 2018). Using
hyperspectral data acquired with diver sensors (space-borne Hyperion, airborne HySpex, and simulated space-borne
EnMAP imagery), spectroradiometric measurements, XRD mineralogical analyses; as well as, applying continuum
removed reflectance spectrum (CRRS), slop and half-area processing methods, the validation of NDGI provides
satisfactory results (Milewski et al., 2019). Coincidentally, the NDGI is simply the SI-$_{ASTER-4,5}$ proposed 16 years
ago by Al-Khaier (2003). Otherwise, we end up with the same index under two different names and different
authors, particularly when the SWIR bands of Landsat sensors or Sentinel-MSI are used, as well as the bands 4 and
5 of ASTER. Obviously, the difference between them is clear when using hyperspectral data.
In other respects, since the emergence of remote sensing as a new scientific discipline in the early 1970s,
vegetation indices (VI's) were involved as radiometric measurements of the spatial and temporal distribution of
vegetation photo-synthetically active. Based on the strong chlorophyll absorption in red and intense reflectivity by
the canopy biomass in NIR, these indices play an important role in deriving various biophysical and physiological
parameters, including percentage of vegetation cover, leaf area index (LAI), absorbed photo-synthetically active





radiation (APAR), production rate of the biomass, etc. Moreover, their interest lies in the detection of changes in
land use and the monitoring of the seasonal dynamics of vegetation on local, regional and global scales (Leeuwen et
al., 1999). Based on the red and NIR bands, the NDVI was proposed by Rouse et al. (1974) at the dawn of remote
sensing. Since these two spectral bands are generally present on Earth observation and meteorological satellites, and
often contain more than 90% of the information relating to vegetation canopy (Baret, 1986; Bannari et al., 1995), the
NDVI had taken a privileged place in the NASA/NOAA Pathfinder project (James and Kalluri, 1994). Thus, it was
daily derived from NOAA-AVHRR data at the Earth scale. Subsequently, it was also derived every day from
MODIS and SPOT-Vegetation data to produce a time series products for global vegetation assessment and
monitoring at the regional and global scales (Chéret and Denux, 2011; Hameid and Bannari, 2016; Liu et al., 2021).
Due to this glorious history and its simplicity, the NDVI has become the most widely used to assess vegetation.
However, despite its popularity and its capability to reduce the sun illumination geometry and to normalize the
topographic variations (Kaufman and Holben, 1993; Bannari et al., 1995), the NDVI shows some sensitivity to the
atmosphere (scattering and absorption) and soil background artefacts (color, brightness, texture, etc.). To overcome
these limitations, more than fifty VI's have been developed and proposed for various applications and under specific
conditions (Bannari et al., 1995). However, despite these new development and innovative efforts, the use of VI's to
characterize vegetation canopy remains limited by various physical factors that affect the recorded signal at the
satellite level, such as atmosphere, sensor-drift, topography, soil background optical properties, saturation, linearity,
and BRDF (Price, 1987; Myneni and Asrar, 1994; Running et al., 1994; Burgess et al., 1995; Bannari et al., 1996;
Teillet et al., 1997; Huete et al., 1997; Bannari et al., 1999).
Cert, the majority of these limiting factors can be corrected on remote sensing imagery or *in situ* measurements
before the extraction of such index; except the impact of the optical properties of the soil background. This last
factor has been considered in the theoretical concept supporting many VI's development for minimising or removing
completely the contribution of the soil underlying the canopy on the remotely sensed signal and, therefore, to
enhance that resulting from the biomass. For instance, the soil adjusted vegetation index (SAVI) was proposed by
Huete (1988) to minimize the artefacts caused by soil background on the estimation of vegetation cover fraction by
incorporating a correction factor "L". Moreover, to overcome the limitations of linearity and saturation, to reduce
the noise of atmospheric effects, and to remove the artefacts of soil optical properties, the enhanced vegetation index
(EVI) was proposed also by Huete *et al.* (2002). Furthermore, the transformed difference vegetation index (TDVI)
was proposed by Bannari *et al.* (2002) to describe the vegetation cover fraction independently to the soil-
background, to reduce the saturation problem, and to enhance the vegetation dynamic range linearly. These indices
(NDVI, SAVI, EVI, and TDVI) were developed and used to establish a close relationship between radiometric
responses and vegetative cover densities. However, despite their particular mission of assessing and managing
vegetation covers, many users of remote sensing applied these indices for soil salinity detection and mapping
(Fernandez-Buces et al., 2006; Aldakheel, 2011; Allbed et al., 2014; Asfaw et al., 2016; Elhag, 2016; Azabdaftari
and Suna, 2016; Ferdous and Rahman, 2017; Neto et al., 2017; Peng et al., 2019; Taghadosi et al., 2019; Nguyen et
al., 2020; Zhu et al., 2021; Golabkesh et al., 2021). Hence the interest of this research to investigate what VI's can
really tell us about the discrimination of soil salinity classes'. The most popular and widely used indices presented



above (NDVI, SAVI, EVI and TDVI) are considered and compared to the newly proposed evaporite mineral indices
(NDGI and SSSI). In this regard, a field survey was conducted for soil and vegetation cover sampling, soil
laboratory analysis, spectral measurements in a Goniometric-Laboratory, and Landsat-OLI image were used. Two
study-sites in arid environments are considered, the Kuwait-State in the Middle-East desert and the Omongwa salt-
pan located in the southwest of Kalahari desert in Namibia.
**2. Materials and Methods**
Fig. 1 summarizes the applied methodology by combining two independent datasets (simulated and image) acquired
over two different study areas located in Kuwait and Namibia. On the Kuwait site, a field survey was conducted and
100 soil samples were collected with various salt contents; as well as a vegetation cover was sampled at different LAI
coverage rates. Then, the bidirectional reflectance factor was measured above each sample of soil and vegetation in a
Goniometric-Laboratory using an Analytical Spectral Device (ASD) spectroradiometer (ASD, 2015). After the
spectral measurements, laboratory analyses of soil samples were achieved to measure the water soluble cations ($Ca^{2+}$,
$Mg^{2+}$, $Na^+$, and $K^+$) and anions ($Cl^-$ and $SO_4^{2-}$) in the extract from saturated soil paste, the pH of saturated soil paste
(pHs) and the electrical conductivity (EC-$_{Lab}$) of the extract from saturated soil paste; as well as the sodium adsorption
ratio (SAR) being calculated using standard calculation procedure (USDA, 2004 and 2014; Zhang et al., 2005). The
results of these analyses provided reliable information on the type and degree of salinity and sodicity in each soil
sample. Thus, they support the interpretation of the complex and close relationship between the soil-salt contents and
their spectroradiometric behaviours. Furthermore, the measured spectra of the most representative soil salinity classes
and LAI densities were transformed using the CRRS (Clark et al., 1987). Likewise, all measured spectra were
resampled and convolved in the solar-reflective spectral bands of OLI sensor using the Canadian Modified Simulation
of a Satellite Signal in the Solar Spectrum (CAM5S) radiative transfer code (Teillet and Santer, 1991), and the relative
spectral response profiles characterizing the OLI sensor filters of each spectral band. Afterwards, the considered
indices were calculated and analysed spectrally, as well as fitted statistically with EC-$_{Lab}$. While, on the Omongwa salt
pan site, the acquired OLI image was pre-processed and converted to the investigated indices. Published by Milewski
et al. (2017), mineralogical ground-truth information collected during previous field work and analysed in the
laboratory, and an accurate Lidar DEM were used for the characterization and validation of the results obtained on
this second site.

210                                                    [ Figure 1 ]

2.1. **Study areas**
The state of Kuwait (Fig. 2) situated in the north western part of the Arabian Peninsula ($29.40^0$ N and $47.50^0$ E) is
characterized by an arid climate, very hot summers ($47\ ^{\circ}C$) and irregular precipitations with an annual mean of 118
mm. The main geomorphological features characterized the study area are escarpments, sand dunes, Sabkhas (pure
salt accumulation), depressions, playas and alluvial fans (Al-Sarawi, 1995). These features are controlled by three
types of surface deposits. The first is represented by Aeolian deposits such as dunes and sand sheet. The second is
identified by evaporites including sodium chloride (halite, NaCl), calcium carbonate (calcite, $CaCO_3$), gypsum





(CaSO$_4$.2H$_2$O), and anhydrite (CaSO$_4$) in coastal and inland Sabkhas. The third include fluvial deposits such as
pebbles and gravels, which are located along the Wadis channels. Each of these deposits has specific geomorphic
characteristics based on their origin, topography that is generally flat with low relief, and climatic impacts.
Geologically, Kuwait stratigraphy consists of two stratigraphic groups; Kuwait Group and Hasa Group (Milton,
1967) consisting of six Formations, four of them are exposed in the outcrops represented by Dammam, Ghar, Mutla
and Jal-AzZor Formations. In Hassa Group, the Dammam Formation *(Eocene)* consists of white fine grained cherty
limestone and forms some karst; however, the three other Formations are composed mostly of sandy limestone,
calcareous sandstones, sand and clay. Soils of Kuwait are mostly categorized as sandy with limited organic matter,
very low nutrient and very high amount of calcareous materials. Moreover, Gatch layer occurs in many Kuwaiti
soils, which is considered a calcic and/or gypsic pan (Milton, 1967).

[ Figure 2 ]

The Omongwa salt-pan area is a natural flat salt playa covering approximately 20 km$^2$ (Fig. 3), located in the south-
west of Kalahari region in Namibia (23°43'S and 19°22'E) at 1200 m altitude above sea level (Genderjahn et al.,
2018). The climate is arid and hot, the average annual temperature is about 20°C with a maximum around 48°C
during the summer (July and August), the average precipitation is about 220 mm/year, and the evaporation exceeds
precipitations. This area is devoid of vegetation except some scattered halophytes in the peripheral neighbourhood
of the north-western of the playa (Milewski et al., 2017). The pan soils are characterized by very low organic matter
content and mixed evaporite sediments (photos in Fig. 3) including halite, gypsum, calcium carbonate, and minor
content of clay (Mees, 1999; Fookes and Lee, 2018; Genderjahn et al., 2018). However, the upper soil surfaces are
mostly dominated by halite crust in variable quantities (Bryant, 1996; Lowenstein and Hardie, 1985), which is
formed over time due to the succession of flooding events in the winter and high temperatures during the summer, as
well as the contribution of wind activity (Schuller et al., 2018; Milewski et al., 2017).

[ Figure 3 ]
**2.1. Soil and vegetation cover sampling**
Soils of Kuwait are mostly sandy with a very low organic matter and are infertile (USDA, 1999). They have been
classified into two main soil orders; the Aridisols occupying 70.8% and the Entisols occupying 23.2% of the area
surveyed, while the other restricted and marginal groups are representing the remaining percentage (6.64%). These
two soil orders are further classified into eight soil great groups based on morphological, mineralogical, chemical
and physical characteristics (Omar and Shahid, 2013; USDA, 1999). The extreme soil salinity class (Sabkhas)
occurs in the Aquisalids soil great group on coastal flats and inland Playas, which contain very high salt contents
and gypsum. High soil salinity class is identified in Haplocalcids that attribute to layer of carbonate masses and salt
contents. Moderate to low salinity class occurs in Petrocalcids soil, which is characterised by calcic hardpan
overlying sandy to loamy soils and presence of scattering halophytes.





Field survey was organized in the center and the east of Kuwait territory (Fig. 2), it includes irrigated
agricultural fields, desert land, urban areas, coastal zones, and low-land such as Bubiyan Island. Based on the
fieldwork and soil map, the following soil salinity classes represented by photos in Fig. 2 were considered: non-
saline (A), low (B), moderate (C), high (D), very high, and extreme salinity (F). The field survey was organized
during four days (15[th] to 18[th] May 2017), and geo-referenced 100 soil samples representing these classes were
collected from upper layer of the soil (0 to 5 cm deep considering an area about $50 \times 50$ cm), placed and numbered
in plastic bags. In addition, each soil sample was physically described (color, brightness, texture, etc.),
photographed, and geographically localized using accurate GPS ($\sigma \leq \pm 30$ cm).
Furthermore, at a medium growth stage, herbaceous vegetation cover canopy (alfalfa and forage plants) with
different LAI coverage rates were collected from the cultivated agricultural fields. A sampling quadrate of 50 cm by
50 cm was used, and all the aboveground biomass (approximately 70 cm height) was harvested within this area. The
samples were immediately stored in bags in a cooler and transported to the laboratory for spectroradiometric
measurements as discussed in the following section 2.3.
**2.2. Soil laboratory analysis**
In the laboratory, the considered soil samples were air-dried, ground, and passed through 2 mm sieve. After the
spectral signatures measurements, the saturated soil paste extract method was utilized to measure the $EC_{-Lab}$ and pH
of saturated soil paste (pHs). Moreover, the major soluble cations ($Ca^{2+}$, $Mg^{2+}$, $Na^+$, and $K^+$) and anions ($Cl^-$ and
$SO_4^{2-}$) were measured, and the sodium adsorption ratio (SAR) was calculated. These analyses have been carried out
at the soil laboratory using methods that meet the current international standards in soil science (Richards, 1954;
Zhang et al., 2005; USDA, 2004 and 2014).
**2.3. Spectroradiometric measurements**
Spectroradiometric measurements were acquired in the Goniometric-Laboratory using an ASD (*Analytical Spectral*
*Devices* Inc., Longmont, CO, USA) FieldSpec-4 Hi-Res (high-resolution) spectroradiometer (ASD, 2015). Equipped
with two detectors with hyperspectral resolution covering the VNIR and SWIR wavelengths (350 and 2500 nm), the
ASD measures a continuous spectrum with a 1.4 nm sampling interval from 350 to 1000 nm and a 2 nm from 1000
to 2500 nm; then it resamples the measurements in 1-nm intervals allowing the acquisition of 2151 contiguous
bands per spectrum. The sensor is characterized by the programming capacity of the integration time, which allows
an increase of the SNR as well as stability.
The bidirectional reflectance spectra were measured above each air-dry soil sample at nadir with a field of view
(FOV) of 25° and a solar (Halogen floodlights) zenith angle of approximately 5° by averaging forty measurements.
The ASD was installed at a height of 60 cm approximately over the target, which makes it possible to observe a
surface of approximately 700 cm². Each soil sample was placed on a black surface to minimise the multiple
scattering effects allowing only the observation and the measurements of the soil signal. For vegetation cover, the
plants were fixed vertically in a black wooden box filled with soil to imitate the *in-situ* canopy at different LAI
coverage's. Similarly to soil samples, the box was placed on a large black surface to minimise the multiple





scattering impacts and only measure the signal reflected by the vegetation canopy. While at this time, the height of
the ASD was about 100 cm over the canopy allowing the observation of a surface with a diameter of 44 cm. A laser
beam was used to locate the center of the ASD-FOV over the center of each target. The reflectance factor of each
sample (soil or vegetation) was calculated by rationing target radiance to the radiance obtained from a calibrated
"Spectralon panel" (Labsphere, 2001) in accordance with the method described by Jackson *et al.* (1980). Moreover,
the corrections were applied for the wavelength dependence and non-lambertian behaviour of the panel (Sandmeier
et al., 1998; ASD, 2015; Ben-Dor et al., 2015).

[ Figure 4 ]

**2.4. Continuum-removal**
Spectral signatures are processed and transformed using numerous approaches to retrieve information regarding the
change in reflectance of particular target over a specific bandwidth between 350 and 2500 nm (Van-Der-Meera,
2004). For instance, absorption features (position, depth, width, and asymmetry) are used to quantitatively estimate
the mineral or chemical composition of samples from the measured spectra in the field, in the laboratory and/or from
hyperspectral images. To emphasize these absorption features, many approaches were proposed including relative
absorption-band-depth (Crowley et al., 1989), spectral feature fitting technique, and Tricorder and Tetracorder
algorithms (Clark et al., 2003). These approaches work on so-called CRRS approach, thus recognizing that the
absorption in a spectrum has a continuum and individual absorption features (Clark *et al.,* 1987; Van-Der-Meera,
2004; Clark *et al.,* 2014). Proposed by Clark and Roush (1984), CRRS transformation and analysis allows the
isolation of individual absorption features in the hyperspectral signature of a specific target under investigation,
analysis and comparison. It normalizes the original spectra and helps to compare individual absorption features from
a common baseline (Clark *et al.,* 1987). The continuum is a convex hull fit over the top of a spectrum under study
using straight-line segments that connect local spectra maxima. The first and last spectral data values are on the hull;
therefore, the first and last bands in the output continuum-removed data file are equal to 1.0. In other words, after
continuum removed, a part of the spectrum without absorption features will have a value of 1, whereas complete
absorption would be near to 0, with most absorptions falling somewhere in between. The CRRS approach was used
for discriminating and mapping rocks and minerals (Clark et al., 1990; Clark and Swayze, 1995), soil salinity
(Farifteh, 2007; Nawar et al., 2014; Bannari et al., 2018; Mousa et al., 2019; Milewski et al., 2019), as well as
vegetation cover (Kokaly et al., 2003; Huang et al., 2004; Manevski et al., 2011). In this study, the continuum
algorithm implemented in ENVI image processing system was used (ENVI, 2012).
**2.5. Spectral sampling and convolving in Landsat-OLI bands**
As discussed above, the measured bidirectional reflectance factors with the ASD have a 1-nm interval allowing the
acquisition of 2151 contiguous hyperspectral bands per spectrum. However, most multispectral remote sensing
sensors measure the reflectance that is integrated over broad bands. Consequently, the measured spectra of each soil
and each vegetation sample was resampled and convolved to match the solar-reflective spectral responses functions



characterizing the optics and electronics of OLI instrument in the VNIR and SWIR spectral bands. In this step, the
resampling procedure considers the nominal width of each spectral band. Then, the convolution process was
executed using the CAM5S radiative transfer code (RTC). This fundamental step simulates the signal received by
OLI sensor at the top of the atmosphere from a surface reflecting solar and sky irradiance at sea level, considering
the filter of each individual band, and assuming ideal atmospheric conditions without scattering or absorption
(Steven et al., 2003; Zhang and Roy, 2016). The reflectance values of soil samples with various salinity degrees and
vegetation cover with different LAI densities were simulated and generated at the satellite-sensor level in VNIR and
SWIR spectral bands of OLI. Thus, the examined VI's and evaporite indices were calculated and statistically
analysed.

**2.6. Landsat-OLI image pre-processing**

Over Omongwa salt-pan site, the used Landsat-OLI image was acquired during the dry season the 28th of September
2016 (Fig. 3) by a very clear day without clouds or cirrus contaminated, and without shadow effects because
topographic variations are absent in this area. Before processing and information extraction, pre-processing
operations have been applied to this image (Teillet et al., 1994; Bannari et al., 1999). Indeed, radiometric sensor-
drift calibration and illumination geometry were corrected to convert the DN to the apparent reflectances at the top
of atmosphere using the irradiance, solar zenith and azimuthal angle values, and absolute calibration parameters
(gain and offset) delivered by USGS-EROS Center in the image metadata file. Thereafter, the atmospheric
interferences measured by the nearest meteorological station to the study site during the acquisition of image were
integrated in CAM5S RTC to simulate and calculate the required atmospheric correction parameters for ground
refelectances retrieval (Pahlevan et al., 2014). The implementation and application of these pre-processing
operations were combined in one-step using PCI-Geomatica (PCI, 2018) to avoid multiple resampling and to
preserve the radiometric integrity of the image data.

**2.7. Spectra and image data processing**

Theoretically, salinity indices (SI) must be highly sensitive to different salinity contents present in the soil surface,
allowing only a qualitative assessment. Nevertheless, they can also be integrated into semi-empirical or physical
models for quantitative prediction of the salinity content classes in the soil (Al-Ali et al., 2021). To select the most
informative soil salinity index, comparative studies have been completed by applying regression analyses between
EC-$_{Lab}$ and SI derived from spectral measurements, satellite, airborne and drone images (Allbed et al., 2014; Bannari
et al., 2018; Peng et al., 2019; Hu et al., 2019; Wei et al., 2020; Milewski et al., 2020; Gopalakrishnan and Kumar,
2020). Often, obtained results vary depending on the spectral wavebands integrated in the equation of each index.
For instance, in irrigated agricultural land in North Africa, the comparison among the SI discussed in the
introduction above pointed out the very limited ability of these indices to differentiate between slight and moderate
salinity classes (Bannari et al., 2016). But they have shown some potential to indicate the impact of soil salinity on
the crop canopy stress. Considering a wide range of salinity contents (from slight to extreme) in arid landscape
(Middle-East) (Shahid et al., 2010), these SI have poorly differentiated the salinity classes (El-Battay et al., 2017,



Bannari et al., 2017b; Al-Ali et al., 2021). On the other hand, these studies showed that the SSSI integrating the
SWIR bands provided the best sensitivity to the presence of salts in the soil. As well as the NDGI (or, SI-$_{ASTER-4,5}$)
performed for evaporite minerals differentiation (Al-Khair, 2003; Milewski et al., 2019). Therefore, these two
indices are considered in the present study and compared to the most popular and widely used VI's (NDVI, SAVI,
EVI and TDVI) to characterize the salinity status in the soil surface. The six indices were implemented and
calculated from simulated data and Landsat-OLI image using EASI-modeling of PCI-Geomatica software (PCI,
2018).


$$\text{NDVI} = \frac{(NIR-R)}{(NIR+R)} \tag{1}$$
$$\text{SAVI} = (1+L) * \frac{(NIR-R)}{(NIR+R+L)} \tag{2}$$
$$\text{EVI} = 2.5 * \frac{(NIR-R)}{(NIR+6R-7.5B+1)} \tag{3}$$
$$\text{TDVI} = 1.5 * \frac{(NIR-R)}{\sqrt{NIR^2+R+0.5}} \tag{4}$$
$$\text{NDGI} = \frac{(SWIR1-SWIR2)}{(SWIR1+SWIR2)} \tag{5}$$
$$\text{SSSI2} = \frac{[(SWIR1*SWIR2)-(SWIR2*SWIR2)]}{SWIR1} \tag{6}$$

Where: R and NIR are the ground reflectance in the red (OLI-4) and near-infrared (OLI-5) spectral bands, "L" is a
correction factor equal 0.5; SWIR1 and SWIR2 are the ground reflectance in shortwave infrared spectral bands,
OLI-6 and OLI-7 bands, respectively.
**3. Results Analysis**
**3.1. Spectral and soil laboratory analyses**
The spectral signatures of the measured 100 soil samples are presented in Fig. 4. These spectra show important
changes in the reflectance's amplitudes and shapes highlighting several absorption features (position and depth). In
the VNIR, they are influenced by several factors including mineralogical composition and assemblage, impurity,
structure, size of salt crystals, and the soil optical properties (color brightness, texture, roughness, etc.). While, in the
SWIR significant absorption features are influenced and controlled by the type and content of the salt mineralogy
existing in each soil sample particularly the gypsum, sodium chloride (halite), calcium carbonate (calcite), and
sodium bicarbonate (nahcolite). Since the impact of moisture content on the measured soil samples is completely



absent or insignificant (0 to 0.05%), only weak absorption bands near 970, 1160, 1350, 1800, and 2208 nm were
observed in some samples (atmospheric water vapor absorption features in 1440 and 1920 nm are not considered in
this analysis).
Furthermore, the EC$_{-Lab}$ revealed that the obtained values are distributed progressively in a wider range between
1.6 and 700 dS.m$^{-1}$, respectively, for agricultural fields and Sabkha "salt scald" consisting of pure salt (halite). These
soil samples present high quantities of chloride (Cl$^-$ : 9.6 to 3932 meq/l), sodium (Na$^+$: 23 to 3615 meq/l),
magnesium (Mg$^{2+}$: 7.8 to 1118 meq/l) and calcium (Ca$^{2+}$: 39 to 230.4 meq/l) than other ions. The dominant ions in
the soil samples are chloride (Cl$^-$) and sodium (Na$^+$) showing, respectively, an R$^2$ of 0.98 and 0.87 with EC$_{-Lab}$.
While, the low relationship occurs with Ca$^{2+}$ (R$^2$ of 0.23) and moderate with Mg$^{2+}$ (R$^2$ of 0.48) and K$^+$ (R$^2$ of 0.46).
The main sources of Cl$^-$ in the soil are from seawater (level rise and spray), precipitation, salt dust, irrigation, and
fertilization. Whereas, parent material, pedogenic processes, irrigation with saline-sodic waters and inappropriate
soil drainage are the main sources of Na$^+$. Likewise, it is observed that the EC$_{-Lab}$ and SAR increased gradually and
very largely from non-saline (EC$_{-Lab}$: 1.6 dS.m$^{-1}$, SAR: 0.4) to extreme salinity in Sabkha (EC$_{-Lab}$: 700 dS.m$^{-1}$, SAR:
445), yielding an R$^2$ of 0.70 between each other. Moreover, the soil pH values ranged from 7 to 7.7 indicated
slightly alkaline reaction due to the presence of bicarbonate (HCO$_3^-$) in the soils with a range from 4 to 10 meq.l$^{-1}$;
as well as, the CaCO$_3$ ranged from 12.5 to 26% showing calcareous soil and parent materials, which significantly
occurs in the arid regions. The results of these chemical analyses showed also the low quantities of organic matter
(OM < 2.6%) in all soil samples, with an average of 0.58%. While the soil texture analysis showed an increase in
salt content with a decrease in soil particle size, which obviously causing significant variation in the amplitude and
the shape of the spectral signatures particularly in the VNIR. Definitely, this spectral confusion masked the effects
of different salt contents in the soil. According to these laboratory analyses, we have a clear idea about the chemical
components and their contents in each soil sample considered in this study.
**3.2. Spectral and CRRS Analysis**
To understand the impact of different salt contents on the spectral behaviour, among the 100 soil samples presented
in Fig. 4 only eight samples are selected with different salinity contents. Their EC$_{-Lab}$ range between 2.4 and 507
dS.m$^{-1}$, pHs between 7.35 to 8.10, and SAR vary from 1.6 to 444.7 (mmoles/L)$^{0.5}$. Fig. 5a illustrates their spectral
signatures noted from A to H, and their characteristics descriptions are summarized in Table 1 (last eight samples in
this table). These spectra show severe confusions in the VNIR regions, which are caused by the soil optical
properties (i.e., color, brightness, texture, etc.) rather than the soil content in the soil. For instance, the reflectance
spectra of sample "D" (195.3 dS.m$^{-1}$) coincide with that of sample "H" (507 dS.m$^{-1}$), although they do not have the
same EC$_{-Lab}$ values, because the soil characteristics play a fundamental role in this confusion (Fig. 5a and Table 1).
In fact, the sample "D" is a sandy soil with small amount of gypsum crystals and shells, and the beginning of salt
crust formation (light gray and white color), while the sample "H" is a pure salt-sabkha (bright florescent halite
crust). Similar confusion is also observed between the opposite samples "A" and "H", respectively, with 2.4 and 507
dS.m$^{-1}$ values of EC$_{-Lab}$. Moreover, the samples "A" and "G" are sandy soils with EC$_{-Lab}$ of 2.4 and 445.5 dS.m$^{-1}$,
respectively; however, they exhibited approximately the same spectral behaviour and amplitude in the VNIR



according to their color (Fig. 5a and Table 1). Consequently, it is impossible to discern or to separate between "D"
and "H" or "A" and "G" samples in the VNIR. This affirmation was also reported by Metternicht and Zinck (1997),
who demonstrated that the soil textures can be a source of spectral confusion between soil salinity classes; as well as
the color and roughness of the soil crusts influenced the reflectance in VNIR and, therefore, causing confusion
among the salts contents in the soil.
On the other hand, the Fig. 5a shows that when the EC-$_{Lab}$ values increase, also the difference among the salt-
affected soil spectra's increase significantly and progressively from 1100 to 2500 nm region of the spectrum. In this
SWIR domain, the spectral signatures of soil samples from "A" to "H" changed progressively in amplitude and
shape according to EC-$_{Lab}$ contents (from 2.4 to 507 dS.m$^{-1}$, see Table 1), as well as a function of SAR (from 1.6 to
444.7 (mmoles/l)$^{0.5}$). The ambiguity between "D" and "H" or "A" and "G" samples observed in the VNIR, is
completely dismissed in the SWIR and it is easy to see gradually the spectral signature position of each sample
according to its EC-$_{Lab}$ content. Definitely, the two SWIR bands of OLI show the highest potential to discriminate
efficiently among different degrees of salinity in the soil (Fig. 5a). These results corroborate those of other
researchers who had shown, for instance, that pure salt (halite, NaCl) does not induce absorption features in the
VNIR (Hunt et al., 1971), and other authors reported some absorption features in SWIR wavebands around 1400,
1900, and 2250 nm (Fig. 5a) that are attributed to dissolved salt in soil moisture and existing liquid in the soil
(Mougenot et al., 1993; Howari et al., 2002a). Moreover, Howari et al. (2002b) and Farifteh (2007) showed that the
depth of absorption features increased with increased salt content in the soil.
[ Table 1 ]

Furthermore, the CRRS transformation of the eight considred soil samples (Table 1) are illustraed in the Fig. 5b. A
total absence of absorption features is observed between 525 and 920 nm, but some features between 350 and 525
nm are revealed. Unfortunately, in this portion of wavelenghts that include the blue band of OLI is not conclusive
because the increase in salinity content does not mean a significant and separate features among soil salinity classes.
Indeed, in this specific electromagnetic window we observe that the sample "H" which is 10 time more saline than
the sample "B" (EC-$_{Lab}$ of 507.0 and 50.5 dS.m$^{-1}$, respectively) are showing similar absorption features. Moreover,
the samples "A", "C" and "E" with different salinity conetnts (EC-$_{Lab}$ of 26.2, 90.0 and 381.0 dS.m$^{-1}$, respectively)
are presenting comparable absorption features (Fig. 5b). This similarty is automatically related to the texture,
raughness, color and brightness of soil samples and not for their salinity content degrees. Infact, "B" and "H"
samples have the same color (white, 10YR 8/1), while the samples "A", "C" and "E" are presenting a very slight
mixt color and brigtness (white-beige, light-gray and light-gray-white) but showing similar Mensel color (10YR 7/2,
Table 1). Moreover, CRRS pointed out that no absorption features charaterizes the salinity in the red and NIR bands
(Fig. 5b). Therefore, this spectral transformation (CRRS) corroborates the original spectral signatures behaviour
which means the impossibility to discriminate among soil salinity classes in VNIR spectral domains. Otherwise,
numerous and significant absorption bands are observed between 920 and 2500 nm highlighting the more suitability
of SWIR wavebands for soil salinity discrmination (i.e., absorption features beyond 950 nm were broadened). In





fact, CRRS has shown that increases in soil salinity (EC-$_{Lab}$) induced automatic changes in the depth of absorption
features, particularly in the water absorption bands, which were shifted toward shorter wavelengths. Consistent
absorption features are observed at wavelengths of 980, 1175, 1448, 1933, and 2430 nm particularly for the pure salt
(sodium chloride) and gypsum samples. These results are in agreement with the findings of Dehaan and Taylor
(2002) and Farifteh (2007).

[ Figure 5 ]


Otherwise, Fig. 6a illustrates the measured spectral signatures of vegetation cover samples at different LAI rates.
These typical spectra of healthy vegetation show the absorption of the visible electromagnetic radiation by the
photosynthetic pigmentation in plants tissues (i.e., carotenoids and chlorophyll). It is well known that each pigment
has different spectral absorption features allowing remote sensing to assess vegetation conditions and, therefore,
give an indication of its overall physiological state (Bannari et al., 2007 and 2008b). The red-edge transition region
between the visible and NIR, from 675 to 750 nm, is informative about vegetation cover diseases and early detection
of pest-attacks (Thenkabail et al., 2018). While in the NIR, a large fraction of the incoming electromagnetic
radiation is reflected toward space according to the biomass density. As we discussed before and also reported by
other studies (Thenkabail et al., 2004; Pacheco et al., 2008), the VNIR spectral domains are the most prominent
regions for green vegetation cover discrimination and the most used in VI's equations. Whereas, in the SWIR
wavelengths, the solar radiation is absorbed by the water content available in the canopy. These wavebands are
indicators of water stress due to water-deficiency in the canopy (Gao et al., 1996; Champagne et al., 2003).
Furthermore, the CCRS emphasizes the wavelengths where significant and gradual changes occurred depending on
the LAI density (Fig. 6b), as well as on the carotene and chlorophyll contents particularly in the blue and red bands
(Fig. 6b). Exceptions occur in the red-edge region (from 675 to 750 nm), and part of NIR spectrum (from 750 to 900
nm), where absorption features are absent. However, at the shorter wavelengths of the NIR from 935 to 1300 nm,
some narrow bands appear characterizing water absorption features. While, stronger absorption features are
observed after 1350 nm (1400-1800 and 1950-2350 nm) due to the variability of internal water content. Therefore,
the subject covered in this section is evident and basically well known by remote sensing community. However, it is
very important to show for the users that the spectral behaviour caused by the internal variability of bio-
physiological parameters (carotenoids, chlorophyll, and water) of vegetation cover is completely different to that
due to the evaporite minerals in VNIR and SWIR wavebands as illustrate by the Figs. 5 and 6.

[ Figure 6 ]

**3.3. Indices validation based on simulated data and measured EC-$_{Lab}$**
In this section, the analysis of VI's capability for soil salinity discrimination was undertaken in two different ways.
The first involves a 2D spectral-space analysis (scatter-plot) relating each index to the reflectance in the red band
(Fig. 7). Among the 100 sampled soils, only 20 samples are considered in this analysis. Their EC-$_{Lab}$ values are





ranging from 2.4 dS.m$^{-1}$ (non-saline soil) to 635 dS.m$^{-1}$ (pure salt, sabkha), and their characteristics are summarized
in Table 1. The 2D spectral-space illustrates how the fraction of vegetation cover is perfectly highlighted by the VI's
(Fig. 7), and predicted correctly and gradually from 50% to 95% proportionally to the increased LAI rates. Whereas
bare soil samples are compressed towards the hypothetical soil-line (Jackson et al., 1983; Huete et al., 1994a and
1994b; Bannari et al., 1996) with null values regardless of their salt content. Indeed, they are quantified by VI's
considering their color in a very limited range values between 0% and 8% for very salty soils with dark and bright
color, respectively. Consequently, undoubtedly VI's cannot exhibit the spatial patterns variability or provide precise
and reliable information about the soil salinity. This finding corroborate those of spectral and CRRS analyses.
Accordingly, if these indices compress and/or eliminate signals coming from the underlying soils, how is it possible
for theme to discriminate the salinity classes, particularly in a large OLI pixel area of 900 m$^2$ with mixt information
of salty soil and vegetation cover fraction?

[ Figure 7 ]


Furthermore, the second part of this analysis considers the totality of 100 soil samples applying a first order
polynomial regression ($p < 0.05$) between the measured EC-$_{Lab}$ and predicted salinity based on the examined VI's
(Fig. 8). Obtained results showed insignificant fits for SAVI, EVI and TDVI ($R^2 \leq 0.06$), as well as for NDVI ($R^2$ of
0.35). Once more, these statistical fits corroborate the spectral signatures and 2D spectral-space analyses, and the
CRRS transformations results that VI's based on VNIR wavebands are not appropriate for correct and accurate
discrimination among various soil salinity classes. Unlike VI's, the evaporite minerals indices have the highest
power for soil salinity discrimination with $R^2$ of 0.71 and 0.72 for NDGI (or SI-$_{ASTER-4,5}$) and SSSI (Figs. 8e and 8f),
respectively. These results are due to the absorption features of salts (gypsum, halite, etc.) in SWIR bands, which are
integrated in the equations of the both indices. Overall, the results are satisfactory and consistent with previous
studies. Indeed, in irrigated agricultural land with slight and moderate salinity, the validation of SSSI derived from
ALI EO-1 with respect to the ground truth showed an excellent fit with $R^2$ of 0.96 (Bannari et al., 2016). Al-Khaier
(2003) showed a good potential of NDGI named also SI-$_{ASTER-4,5}$ ($R^2$ of 0.86) for soil salinity detection in irrigated
agricultural land in Syria using ASTER and ETM+ images. On the basis of spectral measurements of soil samples
collected from Omongwa salt-pan, Milewski et al. (2019) have demonstrated the performance of NDGI for gypsum
content prediction ($R^2$ of 0.84). However, they have shown that this capacity varies with the spatial and spectral
characteristics of the image data used and other sources of problems. Indeed, when NDGI was extracted from
airborne (HySpex) and satellite (Hyperion) hyperspectral images acquired over the pan, the obtained fits showed an
$R^2$ of 0.79 for HySpex with 2.4 m pixel size compared to $R^2$ of 0.71 for Hyperion with 30 m pixel size. Eventually,
this variability can be caused by several problems including the mixture of mineral component fractions within the
pixel size, the low-quality of sensor SNR especially for Hyperian (Kruse, 2001), the residual errors of atmospheric
absorption (Khurshid et al., 2006), the sensitivity of SWIR wavebands to some fragments of senescent vegetation
(i.e., absorption by cellulose and lignin) (Bannari et al., 2015), and the specular effect caused by BRDF problems





(Mishra et al., 2014). However, despite these small variations, the NDGI successfully completed its mission
providing satisfactory results.

[ Figure 8 ]

**3.3. Derived soil salinity maps analysis**
For the interpretation, analysis and validation of the salinity maps derived from Landsat-OLI image acquired over
Omongwa salt-pan site, 14 soil samples collected from the top-surface representing mineralogical ground truth
classes were used (Table 2). These points were sampled and analysed in 2014 and 2015, and published by Milewski
et al. (2017). Moreover, since the soil salinity dynamics occur in response to the way that water moves through and
over the landform following the terrain morphology and topography under the gravity effects (Moore et al., 1993;
Kinthada et al., 2013; Bannari et al., 2021), an accurate Lidar DEM was used (Fig. 9). Generated with a spatial
resolution of 1 m and a vertical accuracy of ±10 cm (Milewski et al., 2017), this DEM undoubtedly supports our
understanding of the topographic impact on the spatial distribution of salinity classes across the pan site. A transects
(A-B) traced from southwest to northeast on the DEM shows the elevation variation between 1227.00 and 1227.80
m with a convex shape and a depth of 80 cm promoting water accumulation, particularly in centre-east and north-
east (Fig. 9).

[ Table 2 ]


[ Figure 9 ]


Fig. 10 illustrates the soil salinity maps derived based on NDVI, SAVI, EVI and TDVI. It is observed that these
indices are blind and unable to detect the presence of salinity in the middle or at the edges of Omongwa pan,
although it is natural flat salt crust playa as shown in Fig. 3. These cartographic products visually corroborate the
results obtained through the analyses of 2D spectral-space (scatter-plots) and CRRS, as well as the statistical fits
with $EC_{-Lab}$. In the center, north and east of the pan some non-classified pixels (black pixels) are due to the absence
of signals that are absorbed by the accumulated water in low topographic areas. Faithful to their mission of detecting
the presence of vegetation, these indices maps are highlighting the presence of scattered halophytes in the peripheral
neighbourhood (north-east and east) of the pan playa. Obviously, this is wrong information about soil salinity
outside the salt-pan. In fact, these results were anticipated because in remote sensing domain it is well known that
the primordial and main mission of VI's is the detection and characterization of photo-synthetically active
vegetation cover as discussed before. Further, they cannot provide any information about the soil because their basic
concept removes the contribution of the soil background from the total signal remotely sensed at the top of
atmosphere as shown in the simulation results. However, in contrast to these results some scientists claim the
predictive power of VI's for soil salinity discrimination and mapping. For instance, Allbed et al. (2014) found that
the SAVI extracted from the IKONOS image is useful for assessing the soil salinity in areas dominated by date palm





trees. On the other hand, when analyzing vegetation cover growth over agricultural lands in South Dakota based on
the time series of MODIS VI, Lobell et al. (2010) observed that EVI was significantly correlated with soil salinity
and more sensitive to changes in salinity stress than NDVI. While, over agricultural soils in California, Whitney et
al. (2018) showed that the temporal interpretation of the time series of MODIS VI's can probably be used to
measure the canopy response to stress caused by soil salinity. Contrary to the conclusions of Lobbel et al. (2010),
Whitney et al. (2018) observed that the strength of the correlation coefficients between VIs and salinity was
generally better for NDVI than for EVI.

[ Figure 10 ]


In the PCI-Geomatica image processing system, the histograms of the derived salinity maps applying SSSI and
NDGI (Fig. 11) were thresholded based on the major salinity classes including non-saline (blue), low (cyan or sky-
blue-green), moderate (clear green), high (yellow), very high (orange-red) and extreme salinity (red-purple). Indeed,
the values of the centroids of the clusters representing these classes were considered; as well as, the standard
deviation value was chosen to limit the overlap between the classes considered and to reduce the chance of a pixel
being classified into more than one class. Fig. 11 shows the spatial distribution of salinity classes across the study
area and in the outer-peripheral regions of the pan. In general, it is observed that the both indices (SSSI and NDGI)
mapped the salinity patterns almost similarly by reflecting the results of the statistical fits discussed above.
However, although the NDGI detects the presence of salinity, it further highlights the gypsum content; particularly
in the borders of the pan (i.e., south, southwest, and north). While, the SSSI further highlights the main salt crusts
present in the pan area that are formed from different mineral sources, including halite, gypsum, calcite, and
sepiolite; as reported ago two decades by Mees (1999) and also recently confirmed by Milewski et al. (2017 and
2019). Moreover, these results are logical since the less soluble carbonates can be found at the edge of the pan,
followed by a succession of sulphates to chlorides towards the central area with lower topography as shown by
DEM (Shaw and Bryant, 2011).

[ Figure 11 ]


Furthermore, the 14 points representing the ground truth (Table 2) are used for the results validation and analysis
process. Their mineralogy is dominated by halite (which appears as white bright and florescent salt crust surface in
Fig. 3), followed by the gypsum as a second most abundant crust (Table 2). Their $EC_{Lab}$ values are ranging between
17.6 and 129.7 $dS.m^{-1}$, and the pH is greater than 8.2 reflecting a strong sodicity coupled with salinity.
Superimposed on salinity maps derived by NDGI and SSSI indices (Fig. 11), most of these points coincide perfectly
with areas showing a high content of salt or gypsum, particularly in the southern borders of the playa. Following the
topographic characteristics of the pan area, a total absence of salinity is observed in the center and center-east parts
of the pan (black pixels) due to the presence of water which absorbs the signal in the SWIR wavelengths. While in
the south and north-western of central part, where the topography is slightly raised, moderate and high salinity



classes are noted by the both indices but more emphasized with SSSI than NDGI. Indeed, because the halite crust covers the majority of pan playa surfaces, inside and outside, with variable contents. Whereas, the gypsum-halite crust mixture surrounded the border as a natural boundary between the interior and the peripheral margin of the pan highlighting a very high to extreme salinity class caused, probably, by the displacement of salt from the surface to the playa edges due to wind action and erosion, as well as by human movements (Bryant, 1996). Moreover, it is also observed that high, very high and extreme salinity classes are associated with slightly high elevation.

According to mineralogical ground truth, the validation points P63, P64, P65 and 172 in the southern region of the pan are dominate by the gypsum crust (33% to 83%) associated with a small amount of halite (5% to 36%). This terrain truth is detected and well mapped by the two indices, but NDGI highlighted more the gypsum belt in south and southwest (Fig. 11). In this region the topography is slightly high and decreases toward the centre-east of the pan, and then it becomes relatively higher in the north and north-west. Points P66 and P67 located on a small circular ridge in the south-central part of the pan with a slight elevation, have almost similar contents of halite and gypsum (45%). However, the salt content in these two points is more stressed in the SSSI map. As well as, nearby points 143 dominated by halite (52%) followed by gypsum (38%) and point 171 with 50% of halite and 27% of quartz, the SSSI map shows more sensitivity to this class than that of NDGI (Fig. 11). The zone surrounding sample point 141 which is nearly pure halite (94%) mixed with very low content of gypsum (3%) is better enhanced by SSSI than NDGI.

Further north of the pan site, areas around the validation points P69, P70 and P71 located at slightly high elevation (~ 1227.8 m) are mapped as very high to extreme salinity classes by SSSI, which also indicates that overall there is an important increase of the salt content in this northern space. These results can be explained by the fact that the SSSI is more sensitive to the halite crust accumulated on the surface which is exposed and clearly visible to the FOV of the Landsat-OLI from space; while the other minerals (less soluble) are precipitated under the layer of halite (Chivas, 2007). The NDGI predict this zone as a moderate salinity class because the mineralization nearby these points indicates the absence or slight content of gypsum and a main mixture of quartz, halite, calcite and sepiolite (Table 2). Indeed, this region was mapped by Milewski et al. (2017) as combined fractions of calcite and sepiolite based on linear spectral mixture analysis (LSMA), hyperspectral imagery and measured endmembers in the field. Nevertheless in the present study, the evaporite indices are applied to the broad bandwidth of the OLI sensor which does not allow the extraction of mineralogical fraction maps like LSMA, but rather a map of salinity showing all salt minerals existing at the surface of study area. However, the results obtained here are very satisfactory and very similar to those obtained by LSMA but all the fractions are combined in one and unique extreme salinity class. Likewise, the results obtained by NDGI in the present study using the OLI image acquired in September 2016 are generally quite similar to those obtained by Milewski et al. (2017). However, only minor differences are observed between the results of these works, because the pan center is heterogeneous and highly dynamic in time (Schuller et al., 2018).

The outer region of the pan, particularly in the east and north, exhibits different salinity classes ranging from moderate to extreme are, probably, associated with wind and dust-storm processes. Indeed, Aeolian salts occur in arid lands consequently through the erosion of salt playa surfaces transported by wind (high concentrations of fine-



grain of salt) and deposited in this area forming sandy-salt-encrusted surfaces. The areas covered by these classes
are certainly located geographically in zones where the wind and sand-storms speed is high. According to
Abuduwaili et al. (2010), the main source of saline dust is the abundance of unconsolidated salt located in enclosed
basins that are affected by strong wind and human disturbance. This type of salinity source has also been widely
observed in the arid Australian landscapes (Zinck and Metternicht, 2009), in the desert of Gobi in China-Mongolia
border region (Wang et al., 2012), in eastern Asia and western Pacific (Zhu and Yang, 2010), in dry playas in the
Mojave Desert, USA (Reynolds et al., 2007), in the shorelines of the Salton Sea in California (Buck et al., 2011),
Aral sea basin-Uzbekistan (Xenarios et al., 2020) and in the deserts of Kuwait (Bannari and Al-Ali, 2020).
Moreover, Aeolian processes were also identified as important salt sediments transport processes in salt playa in
semi-arid south-central of Tunisia (Millington et al., 1989).
**4. Discussion**
The chemical analyses of the 100 examined soil samples disclosed high quantities of chloride ($Cl^-$), sodium ($Na^+$),
magnesium ($Mg^{2+}$) and calcium ($Ca^{2+}$). Nevertheless, the chloride and sodium contents fitted very significantly with
$EC_{-Lab}$, $R^2$ of 0.98 for $Cl^-$ and 0.87 for $Na^+$. It is also revealed that the $EC_{-Lab}$ and SAR values changed progressively
in a wider ranges between non-saline samples ($EC_{-Lab}$ = 1.6 dS.m$^{-1}$, and SAR = 0.4) collected from agricultural fields
and extreme saline soils ($EC_{-Lab}$ = 700 dS.m$^{-1}$, and SAR = 445) sampled from pure salt (halite) and gypsum in
Sabkha. Moreover, the spectral signatures of the considered soil samples illustrate important changes in the
reflectance's amplitudes and shapes (Fig. 4). They revealed severe confusions in the VNIR, which are caused by the
soil optical properties rather than the soil salinity contents. These observations are consistent with the results of
other researchers (Irons et al., 1989; Huete 1989; Metternicht and Zinck, 2003; Bannari et al., 1996 and 2018).
While, the spectra pointed out several absorption features that are linked to the salt mineralogy including gypsum,
halite, calcite and nahcolite, especially in the SWIR wavebands as reported by other scientists (Csillag et al., 1993;
Howari et al., 2002a; Katawatin and Kotrapat, 2005; Farifteh et al., 2008; Mashimbye, 2013; Bannari et al., 2018;
Al-ali et al., 2021). For example, pure halite (NaCl) is transparent and its chemical composition and structure does
not show any absorption features in the VNIR spectral domains, corroborating the finding of Hunt et al. (1971 and
1972). Whereas, absorption bands near the 1420, 1920, and 2250 nm in the spectra of halite are attributed to
moisture and fluid inclusions, as also reported by several authors (Crowley, 1991; Mougenot et al., 1993; Howari et
al., 2002a; Farifteh, 2007).
The CCRS transformations corroborate the trends of spectral signatures and highlighted the confusions in the
VNIR that are caused by the soil optical properties rather than the salt contents in the soil. For instance, very severe
confusion is noted between the soils samples "A" with $EC_{-Lab}$ of 2.4 dS.m$^{-1}$ and "G" with 445.5 dS.m$^{-1}$; despite this
important difference in salt contents they exhibit approximately the same spectral behaviour and amplitude in the
VNIR according to their color and texture (Fig. 5a and Table 1). Consequently, it is impossible to discern or to
separate between soil salinity classes in the VNIR. While when the $EC_{-Lab}$ values increased also the difference
among the salt-affected soil spectra's increased significantly and progressively in the SWIR (Fig. 5a). In this
spectral region, the spectra of soil samples "A" to "H" changed progressively in amplitude and shape due to the



679 increasing values of EC-$_{Lab}$ (from 2.4 to 507 dS.m$^{-1}$). Nevertheless, the noted ambiguity between "A" and "G"

680 samples in the VNIR is completely dismissed in the SWIR. Indeed, it is easy to see gradually the spectral signature

681 position of each sample according to its EC-$_{Lab}$ content; as well as, the CRRS has shown that increases in soil

682 salinity (EC-$_{Lab}$) induced automatic changes in the depth of absorption features. For instance, CRRS analyses of

683 halite-rich soil "H" sample showed consistent absorption features at 960, 1160, 1420, 1780 and 1920 nm and they

684 become deeper, broader, and more asymmetrical with increasing salt content in the soil. Based on several statistical

685 analyses including CRRS, spectral matching techniques, hierarchical classification, and Mann–Whitney U-test;

686 Farifteh (2007) demonstrated rigorously that the SWIR spectral domain contain the most crucial information about

687 soil salinity differentiation. These findings are in agreement with the results of other scientists who characterized

688 several soils rich in sulfates minerals, carbonates and bicarbonates, sodium chloride, etc. (Bowers and Hanks, 1965;

689 Mougenot et al., 1994; Verma et al., 1994; Owen, 1995; Howari et al., 2002a and 2002b; Farifteh et al., 2008; Weng

690 et al., 2008; Masoud, 2014; Nawar et al., 2015; Neto et al., 2017; Bannari et al., 2018; Milewski et al., 2019).

691 Accordingly, the SWIR-1 and SWIR-2 bands of OLI show the highest potential to discriminate efficiently among

692 different degrees of salinity in the soil.

693  Otherwise, spectral analysis and CCRS transformation of LAI at different densities confirmed the relevance of

694 VNIR and SWIR for the assessment of the vegetation cover from viewpoints of biomass, physiological

695 pigmentation and stress. In addition, the 2D spectral-space analysis highlighted the primordial utility of VI's to

696 differentiate the vegetation covers perfectly proportionally to their LAI rates. Unfortunately, the investigated VI's

697 compress the bare soils samples towards the hypothetical soil-line with null values regardless of their salt contents.

698 These irrelevant results are in agreement with those of spectral and CRRS analyses, since VI's are based on blue,

699 red and NIR bands that are not conclusive for soil salinity differentiation. Moreover, statistical regressions ($p <$

700 0.05) between the measured EC-$_{Lab}$ and predicted salinity based on VI's are very insignificant for SAVI, EVI and

701 TDVI ($R^2 \le 0.06$), as well as for NDVI ($R^2$ of 0.35). Certainly, these simulations in an ideal and controlled

702 environment lead to rigorous validation and comparison procedures between the considered indices. In fact,

703 atmospheric interferences are absent, SNR is high, fragments of senescent vegetation are absent, salt contents are

704 well known in soil samples and, consequently, the results obtained are optimal and realistic. These finding are

705 consistant with other results obtained by some researchers who fitted EC-$_{Lab}$ with VI's derived from simulated data,

706 satellite or drone images. For instance, Golabkesh al. (2021) obtained very weak relationship with NDVI ($R^2$ of

707 35%), Ferdous and Rahman (2017) revealed insignificant fit ($R^2 \le 0.03$) with SAVI and NDVI, and Zhang et al.

708 (2011) demonstrated also a low regression trend ($R^2 < 0.28$) for NDVI. These weaknesses are mainly due to the

709 ambiguous information about soil salinity in the VNIR bands. Moreover, based on spectral data and Landsat-OLI

710 image, Al-Ali et al. (2021) demonstrated that the soil salinity models integrating VI's and/or the VNIR bands are

711 inappropriate and inaccurate to predict soil salinity.

712  Over the Omongwa salt-pan site, although the higher albedo of the site centre in the image is principally due to

713 halite crust developed and accumulated during many years as illustrated by true color composite RGB of Landsat-

714 OLI image (Fig. 3), the derived soil salinity maps using VI's are completely unable to detect the slightest grain of

715 salt in the soil. Obviously, the visual analysis of these maps validate and corroborates the previous analyses (i.e.,



spectral, CCRS, 2D spectral-space, and statistical fits) based on simulated data. Obviously, these results were
anticipated knowing that the primordial and main mission of VI's are the detection and characterization of
vegetation canopy by removing the contribution of the soil background from the signal remotely sensed at the top of
atmosphere. Probably, it is possible for VI's to anticipate the vegetation canopy stress caused by the underlying soil
salinity (Lobell et al., 2010; Zhang et al., 2011; Bannari et al., 2016; Whitney et al., 2018), but they do not have the
ability to discriminate and predict soil salinity classes. Definitely, the widespread use of VI's carries inherent risks
misuse by users who exploit remote sensing as a tool, and who have received little or no education in remote sensing
domain (Huang et al., 2020). Indeed, remote sensing is not limited to the story of having an image processing
software packages and free satellite images, but it is a multi-disciplinary and multi-concept scientific fields. It is
based on complex comprehension of a wide range of electromagnetic radiation, reflected or emitted, and its
interaction with the biosphere-atmosphere environment. Hence the interest of this research to investigate whether the
potential of VI's for soil salinity discrimination is a myth or a reality.
Furthermore, considering the simulated data over Kuwait site or the Landsat-OLI image acquired over Omongwa
salt-pan, SWIR wavebands are distinguished by their potential to differentiate among several salt contents in the
soil. The spectral signatures analysis and CRRS transformation showed that increases in soil salinity (EC-$_{Lab}$)
induced automatic changes in the depth of absorption features in SWIR. Statistical regressions between the EC-$_{Lab}$
and evaporite mineral indices showed an excellent and similar discriminating power ($R^2$ of 0.72) for NDGI and SSSI
(Figs. 8e and 8f). Moreover, the salinity maps derived by these two indices illustrate a good spatial distribution of
salinity classes across the study area and in the outer-peripheral regions of the pan (Fig. 11). The both indices
mapped the spatial distribution of salinity patterns almost similarly, corroborating the results obtained from
simulated data and statistical fits discussed above. Overall, the validation of these maps shows a good agreement
with the field truth. However, although the NDGI detects the presence of salinity, it further highlights the gypsum
content; particularly in the borders of the pan (i.e., south, southwest, and north). While, the SSSI further highlights
the main salt crusts present in the pan area that are formed from different mineral sources, including halite, gypsum,
calcite, and sepiolite; as reported by Mees (1999) and confirmed by Milewski et al. (2017 and 2019). In general,
these results are due to the absorption features of salts (gypsum, halite, etc.) in SWIR bands, which are integrated in
the equations of the both indices. Likewise, Al-Ali et al. (2021) showed that the soil salinity models integrating the
SWIR wavebands are the most promising for predicting and quantifying the salt-affected soil classes. Obviously, the
results obtained in the present study are accomplished from Landsat-OLI data but they can also be achieved from
Sentinel-MSI because we demonstrated that these two sensors can be used jointly to monitor accurately the soil
salinity and it's dynamic in time and space in arid landscape (Bannari et al., 2020).
**5. Conclusions**
In the present study, we analyzed the potential and limits of vegetation indices compared to evaporite mineral
indices for soil salinity discrimination and mapping in arid landscapes. To achieve these, combined approaches that
exploit simulated spectral data and OLI image acquired over two study sites were used. The first site is the Kuwait-
State in Middle-East and the second site is the Omongwa salt-pan in Namibia. Field survey was organized and 100



soil samples with various salt contents were collected, as well as samples of vegetation covers with different LAI
densities. Spectroradiometric measurements were acquired in a Goniometric-Laboratory above the soil and
vegetation samples using the ASD spectrometer. To understand the complexity of the close relationship between the
salt contents in the soil and their spectral signatures, soil chemical analyses were accomplished. Indeed, soluble
cations and anions, pH and EC$_{-Lab}$ were measured, and the SAR was calculated. Furthermore, the spectra of the most
representative soil salinity classes and LAI densities were transformed using the CRRS. Likewise, all measured
spectra were resampled and convolved in the solar-reflective spectral bands of OLI sensor. Afterwards, the indices
were calculated and analysed in 2D spectral-space, and fitted statistically ($p < 0.05$) with EC$_{-Lab}$. Moreover, the
acquired OLI image over Omongwa salt pan site was pre-processed and converted to the considered indices.
Accurate Lidar DEM was used to support visual analysis and interpretation, as well as mineralogical ground-truth
information collected and analysed previously were considered for the characterization and validation of the derived
maps on this second site.
The results show that the soil spectral signatures are very sensitive to soil salinity contents. Their shapes, forms
and amplitudes changed gradually depending on the salt contents (EC$_{-Lab}$). According to chemical soil laboratory
analyses, the measured amounts of EC$_{-Lab}$ in the examined soil samples are due to the chloride, sodium, magnesium
and calcium which pointed out several absorption features in the spectra, particularly in the SWIR wavebands. On
the other hand, it is also observed that the soil optical properties (color, brightness, texture, roughness, etc.) have an
impact on these spectra, especially in the VNIR spectral domains. Overall, spectral analysis and CCRS
transformation highlighted severe confusions of soil salinity classes in the VNIR wavelengths due to soil artefacts
rather than the salt contents. Moreover, they revealed that blue band integrated in EVI equation is inconclusive for
soil salinity differentiation, and the salts minerals absorption features are completely absent in the red and NIR
bands that are generally used by VI's. While the SWIR wavebands show the highest potential for efficient
discrimination among soil salinity classes.
Furthermore, spectral signatures analysis and CRRS transformation showed that the VNIR and SWIR fulfill their
essential conditions to be sensitive to vegetation cover density and its physiological constituents. In addition, 2D
spectral-space investigation results highlighted the primordial utility of VI's to differentiate the vegetation covers
perfectly and proportionally to rates of LAI. While, regardless the salt contents in the soil samples, VI's are not
conclusive as their fundamental concept eliminates the underlying soil contribution on the remotely sensed signal.
These unsuccessful results are corroborated by statistical fits ($p \leq 0.05$), between the measured EC$_{-Lab}$ and VI's, who
achieved very low coefficients of determination, $R^2 \leq 0.06$ for SAVI, EVI, and TDVI, and $R^2$ of 0.35 for NDVI.
Likewise, although the higher albedo of Omongwa salt-pan site due to halite crust developed and accumulated over
years, the soil salinity maps derived from OLI image based on VI's are completely unable to detect the slightest
grain of salt in the soil. Overall, regardless the data used, the processing method, the study site and the validation
procedures, the results obtained converge toward the same conclusions that it is impossible for VI's to detect the
spatial patterns variability or to provide precise and reliable information about the soil salinity classes.
Finally, considering the simulated data over Kuwait site, statistical regressions between the measured EC$_{-Lab}$ and
the evaporite mineral indices showed significant discriminating power ($R^2$ of 0.72) for NDGI and SSSI. Moreover,



the derived maps from the OLI image based on these two indices over Omongwa salt-pan surface illustrated a good spatial distribution of salinity classes across and in the outer-peripheral regions of the pan site. Overall, the validation of these maps shows an excellent agreement with the field truth. However, although the NDGI detects the presence of salinity, it highlights the gypsum content; particularly in borders of the pan (i.e., south, southwest, and north). While, the SSSI further accentuates the main salt crusts present in the pan area that are formed from different mineral sources, including halite, gypsum, calcite, and sepiolite. In general, these results are due to the absorption features of gypsum and halite in SWIR bands, which are integrated in the equations of the both indices. Accordingly, NDGI and SSSI can be used to predict and monitor soil salinity and its dynamics in time and space in arid landscapes.

**6. Author Contributions:** Professor A.Bannari performed the paper conceptualization, data collection, pre-processing and processing, results analyses and paper writing. Professor A. Abuelgasim assisted in the results analyses and the paper writing. All authors have read and agreed to the published version of the manuscript.

**7. Competing Interests:** The authors declare no conflict of interest.

**8. Acknowledgments**
The authors would like to thank Dr. Shabbir A. Shahid (soil scientist specialist) from Kuwait Institute for Scientific Research (KISR) for the assistance during the soil laboratory analyses and for the review of this paper. They acknowledge the NASA-USGS for Landsat data and they express their gratitude to the anonymous reviewers for their constructive comments.

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

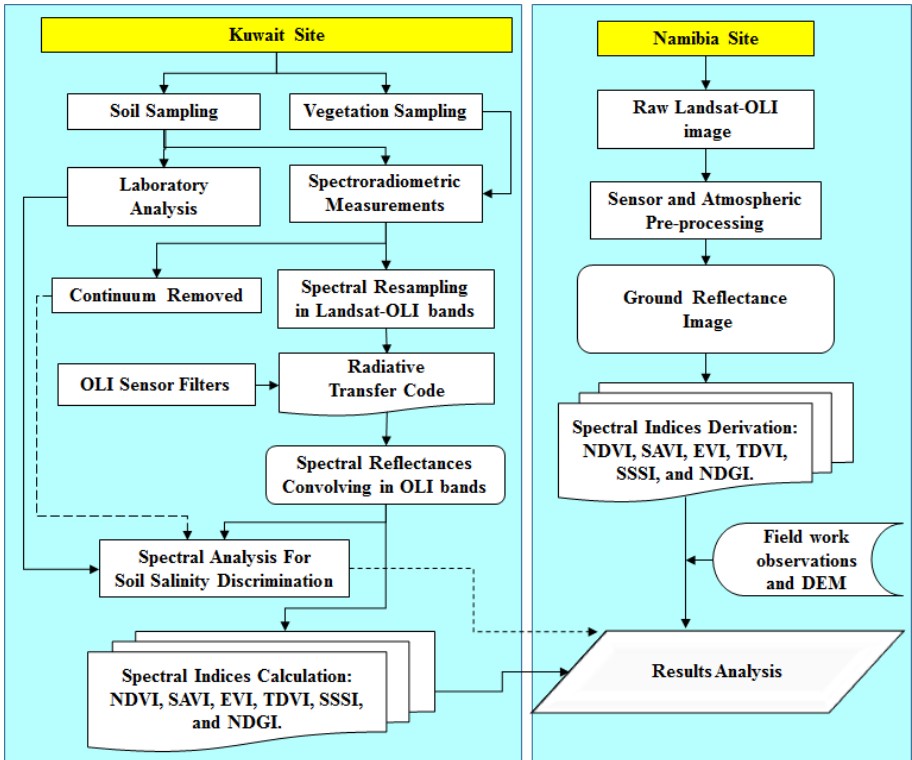


**Figure 1:** Methodology Flowchart.














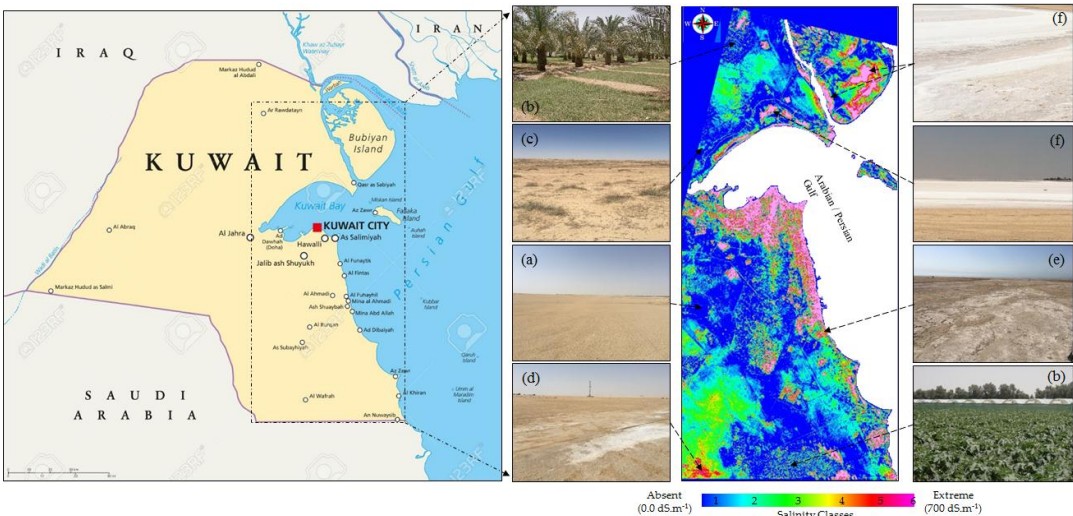


**Figure 2:** Maps of Kuwait study site location and soil salinity with photos illustrating salinity classes (Maps and photos from: Bannari and Al-Ali, 2020).


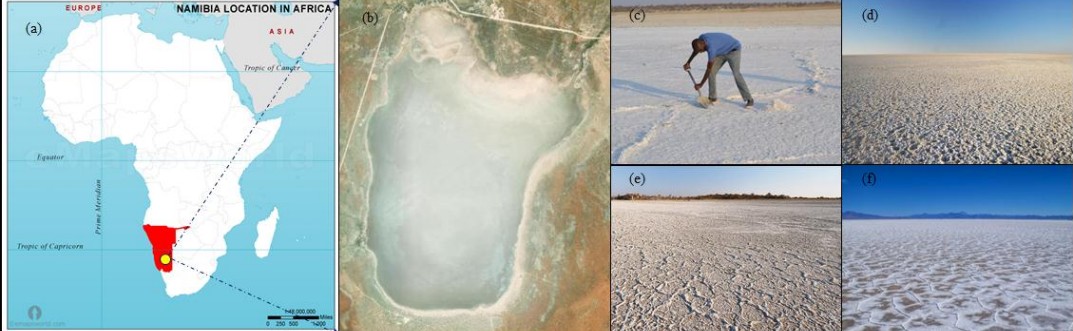


**Figure 3.** Location map of Omongwa salt playa site in Namibia in Africa (a, from Esri), Landsat-OLI RGB image (b), and photos (c-f) illustrating the accumulation of salt crust (source: https://www.namibiansun.com/news/gecko-denies-legal-threat2017-12-01).











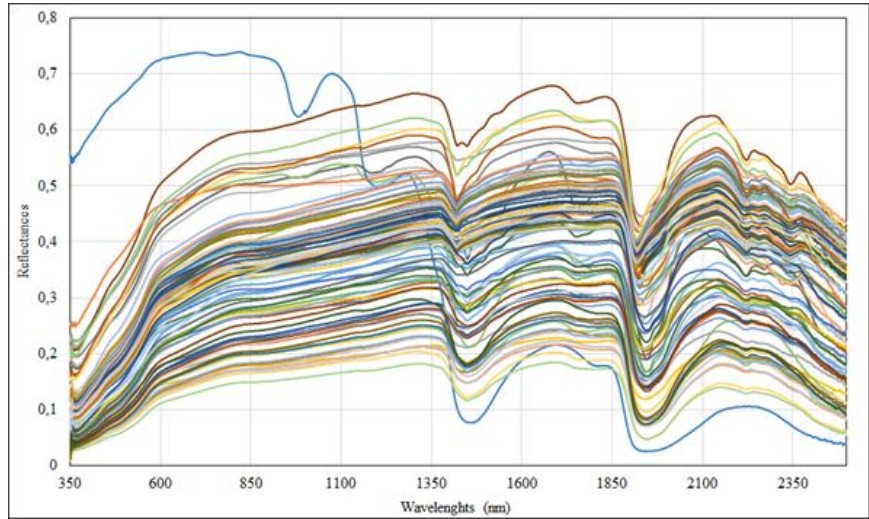


**Figure 4.** Spectral signatures of 100 soil samples with different degrees of salinity.



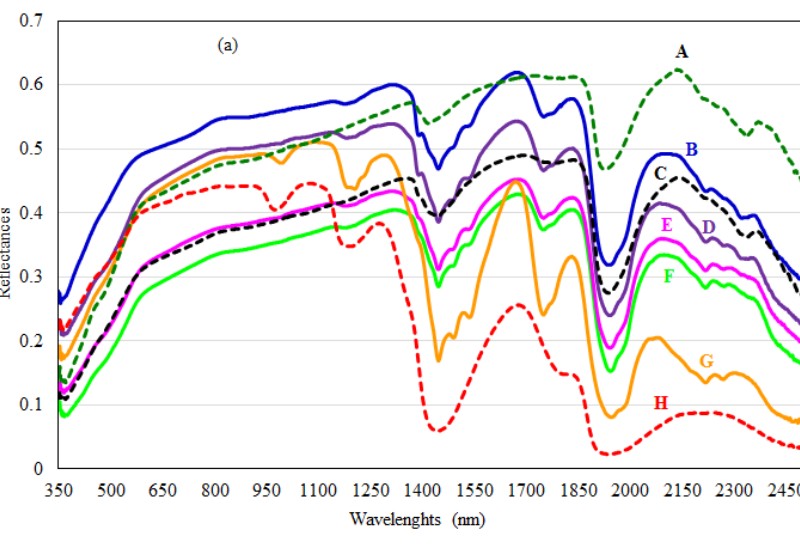


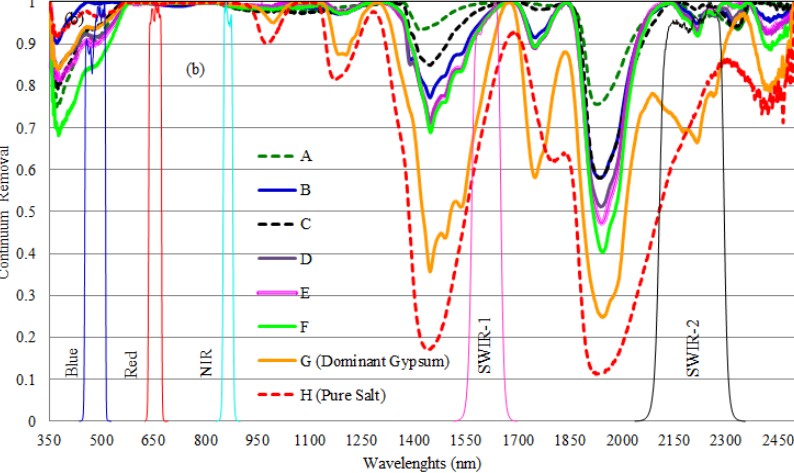


**Figure 5.** Spectral signatures of some soil samples with different salt contents (a) and their continuum removal with
OLI filters profiles in blue, red, NIR and SWIR bands (b).







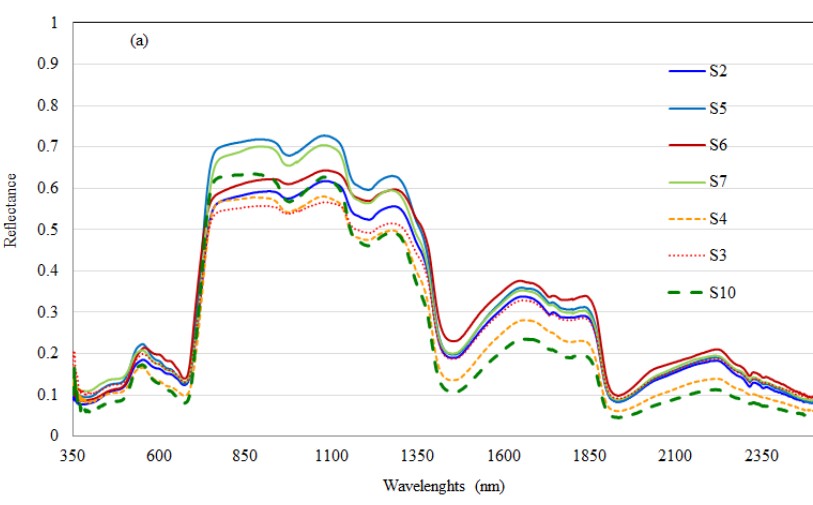


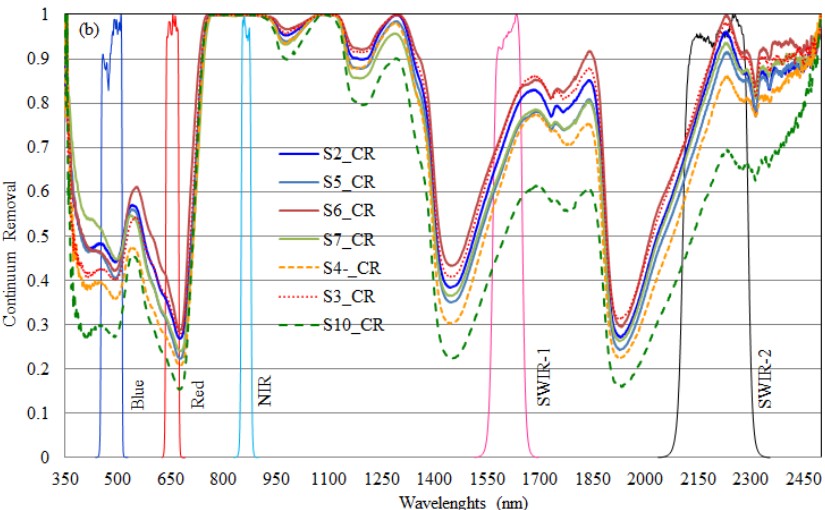


**Figure 6.** Spectral signatures of vegetation cover with different LAI (a) and their continuum removal with OLI
filters profiles in blue, red, NIR and SWIR bands (b).












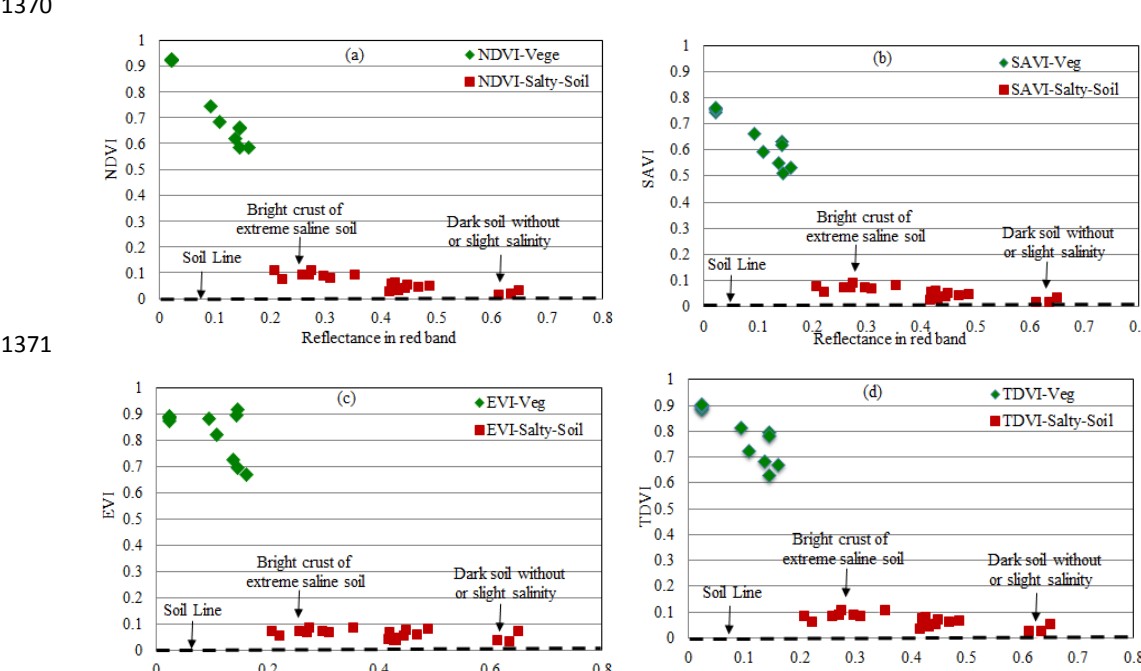



**Figure 7.** Sensitivity of VI's to discriminate soil salinity contents and LAI rates.

















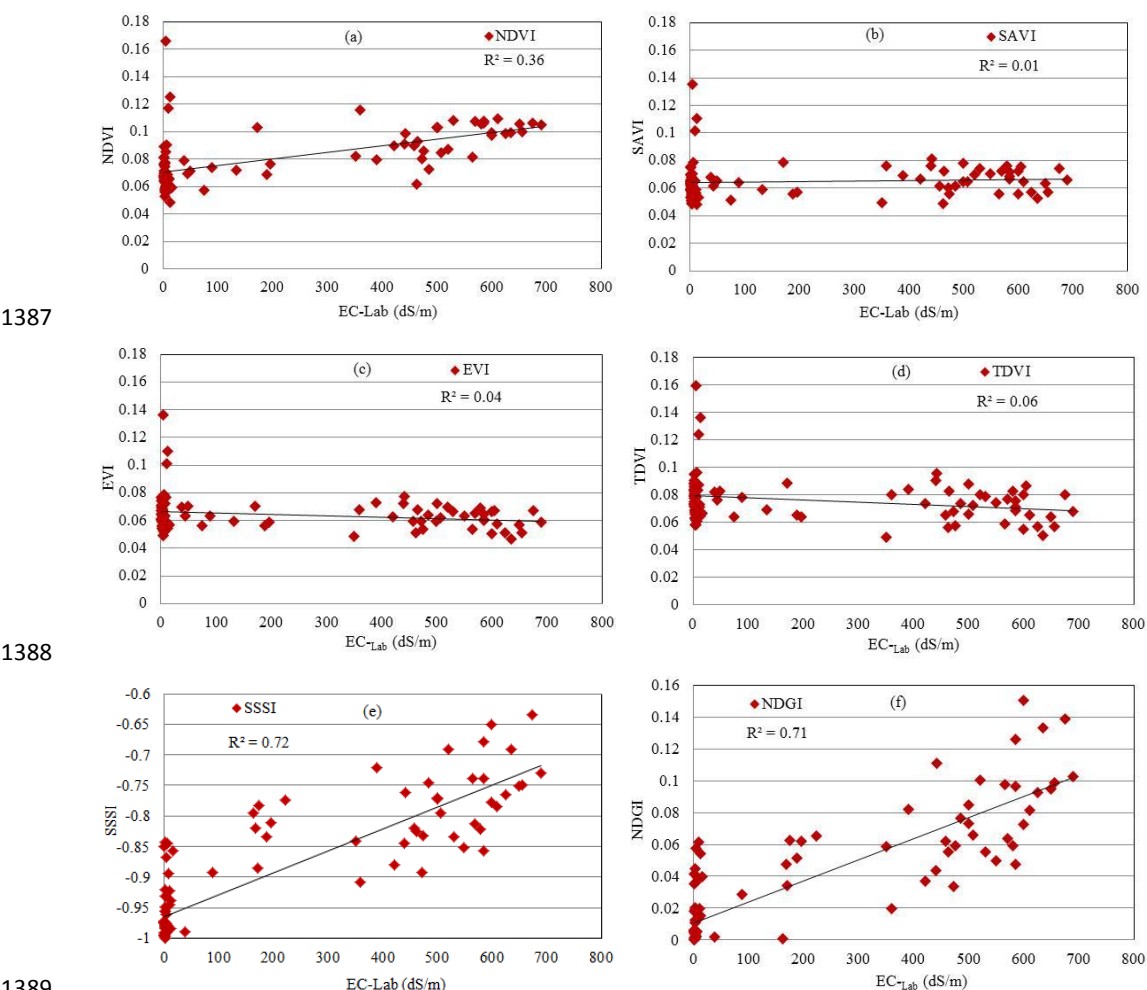




**Figure 8.** Statistical fits between the considered VI's and EC-Lab.





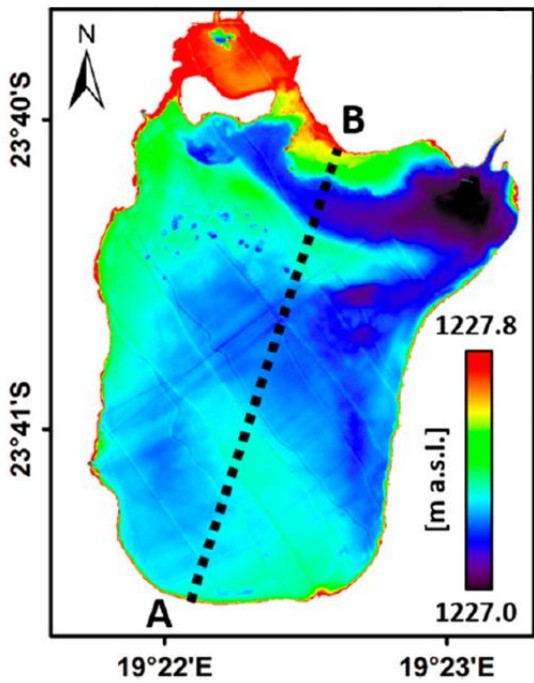


**Figure 9.** Lidar DEM characterizing the topographic variability across the pan site (Milewski et al., 2017).






**Figure 10.** Assumed soil salinity maps derived based on VI's.




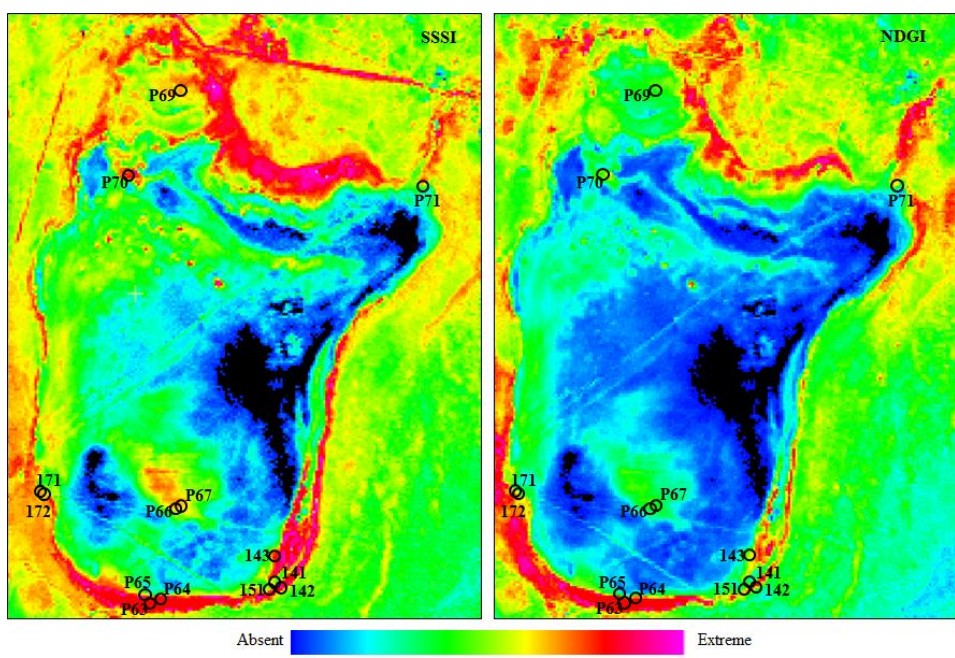

**Figure 11.** Soil salinity maps derived by evaporite mineral indices: SSSI and NDGI.





**Table 1.** Description of some considered soil samples.

| Sample Number | Sample Name | Munsell Color | EC (dS.m$^{-1}$) | pHs | SAR (mmoles/L)$^{0.5}$ | Texture | Description |
|---|---|---|---|---|---|---|---|
| 1 | K-19 | 7.5 YR | 18.2 | 6.77 | 30.20 | Sandy | Sandy soil without gypsum and shells |
| 2 | K-84 | 10 YR | 635.0 | 6.24 | 449.50 | Sandy-Clay-Loam | Crust of salt with small amount of gypsum |
| 3 | K-88 | 10 YR | 583.0 | 6.46 | 468.10 | Sandy-Clay-Loam | Crust of salt with small amount of gypsum |
| 4 | S-10-1 | 10YR 7/2 | 247.5 | 8.85 | 309.26 | Sandy-Loam | Dominant salt crust with gypsum crystals |
| 5 | S-41-1 | 10YR 8/1 | 506.9 | 6.90 | 425.00 | Pure salt (halit) | Salt scald-Sabkha |
| 6 | S-49-2 | 10YR 7/3 | 108.6 | 7.70 | 256.00 | Sandy | Sandy saline soil without gypsum |
| 7 | Ba-33-B | 18YR 8/1 | 7.3 | 8.16 | 57.79 | Loamy-Sandy | Sandy soil with slight salinity, agricultural farm |
| 8 | Ba-33-C | 5Y 8/1 | 5.5 | 8.39 | 55.13 | Loamy-Sandy | Sandy soil with slight salinity, agricultural farm |
| 9 | Ba-15-C | 10YR 8/1 | 399.0 | 7.71 | 298.85 | Sandy-Loam | Pure gypsum crystal deposited by wind erosion |
| 10 | Ba-16-C | 10YR 7/1 | 388.0 | 7.61 | 403.56 | Sandy-Loam | Gypsum rocks at the surface |
| 11 | Pure-salt-Dry | 10YR 8/1 | 509.0 | 7.50 | 465.00 | Pure salt (halite) | Salt scald-Sabkha |
| 12 | Pure-Salt-Wet | 10YR 8.1 | 512.0 | 7.10 | 389.00 | Pure salt (halite) | Salt scald-Sabkha |
| 13 | A | 10YR 7/6 | 2.4 | 7.7 | 1.60 | Sandy | Sandy soil without gypsum and shells |
| 14 | B | 10YR 8/1 | 55.6 | 8.10 | 84.50 | Sandy-Clay-Loam | With small amount of gypsum crystals and shells |
| 15 | C | 10YR 7/2 | 119.6 | 7.71 | 162.00 | Loamy-Sandy | Sandy soil with small amount of gypsum crystals and shells |
| 16 | D | 10YR 7/2 | 195.3 | 7.47 | 225.90 | Sandy-Loam | Beginning of salt crust formation. Small amount of gypsum crystals and shells |
| 17 | E | 10YR 7/2 | 333.0 | 7.57 | 325.20 | Sandy-Clay-Loam | Beginning of salt crust formation. Small amount of gypsum crystals and shells |
| 18 | F | 10YR 7/2 | 406.5 | 7.35 | 403.60 | Sandy-Clay-Loam | Crust of salt with gypsum, calcium carbonate, and small amount of shells |
| 19 | G | 5Y 8/1 | 445.5 | 7.60 | 415.20 | Sandy | Mixture of pure gypsum crystal and salt deposited by wind erosion |
| 20 | H | 10YR 8/1 | 507.0 | 7.60 | 444.70 | Pure salt (halite) | Salt scald-Sabkha |










**Table 2.** Laboratory analyses of Omongwa soil surface samples (locations in Fig. 11).

| Point Number | EC-$_{Lab}$ (dS.m$^{-1}$) | pHs | Mineralogy analysis (in %) | | | | | Soil |
|---|---|---|---|---|---|---|---|---|
| | | | Halite | Gypsum | Quartz | Calcite | Sepiolite | |
| P63 | 17.6 | 8.6 | 5 | 75 | 19 | 0 | 0 | Sandy |
| P64 | 17.6 | 8.6 | 36 | 47 | 17 | - | - | - |
| P65 | 42.3 | 8.5 | 14 | 83 | 3 | - | - | - |
| P66 | 42.3 | 8.5 | 41 | 44 | 13 | - | 1 | - |
| P67 | 80.7 | 8.4 | 46 | 45 | 3 | - | 5 | Sandy and Silty |
| P69 | 36.5 | 8.7 | 15 | 0 | 28 | 41 | 15 | - |
| P70 | 33.7 | 8.6 | 16 | 27 | 32 | 15 | 11 | - |
| P71 | 33.7 | 8.6 | 7 | 0 | 26 | 47 | 6 | - |
| 141 | 33.7 | 8.6 | 94 | 3 | 1 | 0 | 0 | - |
| 142 | 23.4 | 8.8 | 9 | 15 | 63 | 7 | 5 | - |
| 143 | 129.7 | 8.3 | 52 | 38 | 1 | 4 | 5 | Silty and Sandy |
| 151 | 48.2 | 9.2 | 21 | 7 | 64 | 8 | 0 | Sandy and Silty |
| 171 | 98.0 | 9.0 | 50 | 17 | 27 | 4 | - | - |
| 172 | 42.3 | 8.7 | 17 | 33 | 41 | 8 | - | - |

These laboratory analyses results are adapted from Milewski et al. (2017).