# Peer review of "Potential and limits of vegetation indices compared to evaporite mineral indices for soil salinity discrimination and mapping"

_SOIL, 2021_

## Referee Comment (RC1)

**OLI - SSSI**

**OLI - RGB**

**OLI - spectra**

---

## Community Comment (CC1)

The manuscript, first of all, parameters related to salinity in soil samples in Kuwait and their relationships with groun-based hyperspectral data are examined. However, hyperspectral data obtained in Kuwait is "resampled" to Landsat 8 OLI bands. Information from here is used to predict soil salinity at Omangwa salt-pan in Namibia.

The study differentiates itself by presenting an approach towards the applicability of the modeling information obtained in similar areas.

**General comment**

Assertively, modeling approaches from vegetation indexes are reported to have failed at Omongwa salt pan. This is an expected result. In addition, not only landsat, but any satellite image that is currently accessible for free and paid cannot detect a salt crystal due to spatial resolution.

We can create the polynomial model in Microsoft excel. When applying this model spatially in a raster environment with commercial software such as PCI, it becomes difficult to understand in text. At least I struggled.

**Specific comments and technical corrections**

A detailed literature review on the subject of the study, in which the literature is abundant, is given.

Section 2.3, L283-285

I calculated this part myself.

$$r = h * \tan((\frac{\alpha}{2}) * \frac{\Pi}{180}) \tag{1}$$

$$A = \Pi * r^2 \tag{2}$$

α, the measuring wieving angle

A, Field of view

r, Field of view radius

You know that it is calculated using Equations 1 and 2.

According to your data, when I calculate it, I get 554.99 cm2. Saying about 700 would lose precision.

You made the resampling function in Section 2.5., L326 using "CAM5S radiative transfer code (RTC)". Was this process performed in an open source software or with a commercial software? Readers may want to know about it. There is one that I know of. The "hsdar" package can do these things in R.

hsdar: Manage, Analyze and Simulate Hyperspectral Data. https://cran.r-project.org/web/packages/hsdar/index.html

L1301,Figure 1, It's really hard to follow the arrows in the flowchart.

L1420, Table 1 and Table 2, Wouldn't it be more appropriate to give descriptive statistics in general terms? For example, it may be important to know the salt content of the soils taken from the A, B, C, D, E and F regions in Table 1 Kuwait.

---

## Author Comment (AC1)

**Answers for CC1**

**General comment**

Assertively, modeling approaches from vegetation indexes are reported to have failed at Omongwa salt pan. This is an expected result. In addition, not only landsat, but any satellite image that is currently accessible for free and paid cannot detect a salt crystal due to spatial resolution.

Answer:
Yes, this result was expected, and this is the reason way we are showing for remote sensing users that vegetation indices are not appropriate for soil salinity mapping. Indeed, the reasons behind this paper is to show for people who use remote sensing as a tool, and who have received little or no education in remote sensing domain, that vegetation indices do not have the ability to discriminate and predict soil salinity classes.

We can create the polynomial model in Microsoft excel. When applying this model spatially in a raster environment with commercial software such as PCI, it becomes difficult to understand in text. At least I struggled.

Answer:
Many models and algorithms exist in the literature. The semi-empirical models used in this study are based on spectral indices, easier to transfer between sensors and can be used as a robust alternative compared to the revolutionary and complex modelling methods. They are based on the knowledge of spectral absorption features that characterize specifically the target under investigation. They were developed based on modeling, laboratory analysis and spectral measurements in Excel or Matlab environments. Then, they were applied and validated using image processing system such as PCI, ENVI or ERDAS. This process is well known and very easy to understand by the remote sensing scientific community.

**Specific comments and technical corrections**

A detailed literature review on the subject of the study, in which the literature is abundant, is given.

Section 2.3, L283-285

I calculated this part myself.

α, the measuring wieving angle

A, Field of view

r, Field of view radius

According to your data, when I calculate it, I get 554.99 cm2. Saying about 700 would lose precision.

Answer:
Your calculation is valid for a circle measurement in 2D, and some people who us remote sensing as a tool apply this equation to have an idea about the observed surface. However,

the remote sensing measurements are acquired in 3D dimensions, following a solid in W/m²/μm/Sr. The measured electromagnetic energy (EME) pass through a specific FOV and recorded by CCD array, which is not circular but a linear array. Obviously, the ASD simulate the satellite measurements that record the reflected EME by the CCD array following the across-track or along-track modes that are not circular. For instance, the acquired images by Landsat (MSS, TM, ETM+, and OLI), Sentinel, etc. are not circular. Please, for more details see the ASD manual: http://www.gep.uchile.cl/Biblioteca/radiometr%C3%ADa%20de%20campo/TechGuide.pdf.

You made the resampling function in Section 2.5., L326 using "CAM5S radiative transfer code (RTC)". Was this process performed in an open source software or with a commercial software? Readers may want to know about it. There is one that I know of. The "hsdar" package can do these things in R.  **hsdar**: Manage, Analyze and Simulate Hyperspectral Data. https://cran.r-project.org/web/packages/hsdar/index.html

For multispectral remote sensing, the professional and most used RTC by NASA, ESA, CSA, etc., are 5S (Tanre et al., 1990), CAM5S (Teillet and Santer, 1991), 6S (Vermote et al., 1997). MODTRAN (Berk et al., 1989) versions are also used in Hyperspectral.

- Tanre, D., Deroo, C, Duhaut, P., Herman, M., Morcrette, J.1., Perbos, J. and Deschamps, P.Y. (1990). *Simulation of the Satellite Signal in the Solar Spectrum (5S).* Laboratoire d'Optique Atmospherique, Universite des Sciences et Techniques de Lille, Lille, France, 320 pages.
- Teillet, P.M., and Santer, R.: Terrain Elevation and Sensor Altitude Dependence in a Semi-Analytical Atmospheric Code". Canadian J. of Remote Sens. 17, 36-44, 1991.
- Vermote, E., Tanre, D., Deuze, J.L. et al. (1997) Second simulation of the satellite signal in the solar spectrum, 6S: An overview," IEEE Transactions on Geo-science and Remote Sensing, 1997, 35(3): 675-686.
- Berk, A., Bernstein, L.S., and Robertson, D.C. 1989. *MODTRAN: a moderate resolution model for LOWTRAN 7.* Air Force Geophysics Laboratory (AFGL), Hanscom Air Force Base, Md., Final Report GL-TR-0122. 42 pp.

L1301,Figure 1, It's really hard to follow the arrows in the flowchart.

Answer:
After revision, we find that the flowchart is clear and easy to follow.

L1420, Table 1 and Table 2, Wouldn't it be more appropriate to give descriptive statistics in general terms? For example, it may be important to know the salt content of the soils taken from the A, B, C, D, E and F regions in Table 1 Kuwait.

Answer:
Descriptive statistics are not relevant for this research analyses.

**Dear Sir many thanks for your interest to our research**

Professor A. Bannari
https://www.researchgate.net/profile/Bannari-Abdou
https://scholar.google.fr/citations?user=D6sZhSQAAAAJ&hl=fr

---

## Author Comment (AC2)

**General comments**

This research tests common Landsat 8 OLI multispectral vegetation indices and band ratios based on the SWIR information for soil salinity mapping and tests this at two salt rich, arid environments, one coastal region in Kuwait and a salt pan in Namibia. With the topic of soil salinity mapping a relevant scientific question is addressed in this study. The overall pre-print and is fairly well structured and written, although results and discussion should be separated more clearly. However, relevant concerns relate to the overall research design, the applicability of the methods used, and the novelty of the work presented. Firstly, the sensitivity of spectral vegetation indices is tested for surfaces that are vegetation free. Obviously, the result shows that vegetation indices and soil salinity have no correlation in such a vegetation free environment. For other (at least sparsely vegetated) areas with slightly to moderate saline soils such an approach might be relevant for the (indirect) mapping of soil salinity by the proxy of vegetation vitality/density. The potential and limitation of such a concept would be worth studying, however, at several parts in the pre-print the authors make clear that the objective is to test vegetation indices for salinity mapping of none-vegetated surfaces, which has no theoretical basis at all and a test seems not useful. Considering this study design, the objective on "analyzed the potential and limits of vegetation indices compared to evaporite evaporite mineral indices for soil salinity discrimination and mapping in arid landscapes" cannot be answered fairly. Secondly, the SWIR band index termed SSSI2 is already tested in numerous previous studies with contribution of the author, e.g., Bannari et al. (2008a), Bannari et al. (2017b), El-Battay et al. (2017), Bannari & Al-Ali (2020). In Bannari & Al-Ali (2020) even uses the same Kuwait data that is exploited in this study. The only difference is that the Bannari & Al-Ali (2020) study explores actual Landsat 8 OLI data, whereas in this study the OLI spectra a simulated from laboratory spectra (without explaining the reasoning), which arguably produce results that are not as relevant compared to real spaceborne data. Thirdly, the second SWIR band index termed NDGI seems to be completely misused in the frame of this study. Originally proposed in Milewski et al. (2019) as a narrow band index focusing on the estimation of gypsum content, it is without much explanation adapted as broadband Landsat OLI index for general salinity mapping, although the OLI bands do not even cover the same wavelength. In this "adapted" form the NDGI contains basically the same information than the SSSI2 index (OLI SWIR bands) only in slightly different formula. Finally, the relationship between SWIR indices and specific salt mineralogy is not discussed regarding the impacts to the results and misinterpretations of the surface salinity are the consequence.

In summary, 1) the most relevant part of the proposed methodology (SSSI2 index) is already tested in several studies by the author, 2) vegetation indices are tested for soil salinity detection of vegetation free surfaces and 3) an additional spectral index (narrow band) seems misused. These raised points question the overall relevance and/or novelty of the presented work. From this reviewer's perspective the submission should be rejected mainly as research design is not able to fairly address the research question and the novelty of the remaining results seems minor.

Answer:
Dear reviewer, thank you for your time, suggestions and comments regarding our paper. However, I would like to remember you that the reasons behind this paper is to show for people who exploit remote sensing as a tool, and who have received little or no education in remote sensing domain, that vegetation indices do not have the ability to discriminate

and predict soil salinity classes. Second, during the last two decades many scientist demonstrated that the SWIR region is the most relevant for soil salinity discrimination. In our papers during the last decade (that you are mentioning above), we demonstrated the utility of these wavelengths for soil salinity mapping in many environment and using different satellites data (TM, ETM+, ALI, OLI, Sentinel-MSI and Worldview). We want to highlight the importance of this concept and to show that vegetation indices based on the visible and NIR bands are not appropriate for soil salinity discrimination. Third, the NDGI was not developed only for narrow bands and was not misused in the frame of this study, **this statement is NOT CORRECT**. Indeed, always in remote sensing domain, we analyze the spectral signatures to find the absorption features of the target under investigation and, then, we apply them in the modeling process using hyperspectral or multispectral data. Obviously, with the condition that the proposed features must be covered by the used bands of the used sensor. Moreover, I would like to attract you attention that Milewski et al. (2020) used the NDGI for Landsat-8 data which are broad bands (please, see the following publication). In general, keep in mind that an index can be applied using narrow or broad bands once the exploited band covers the absorption features. Certainly, the narrow bands highlight mineral classes better than broad bands. Finally, I would like to highlight that according to my 35 years' experience in remote sensing domain and soil salinity mapping that this paper is original and pragmatic. Without a doubt, this work will be a reference for users of remote sensing as a tool. Indeed, during its exposition in researchgate during two months, it was recommended by many people and read by 110 scientists.

Milewski, R., Chabrillat, S., and Bookhagen, B.: Analyses of Namibian Seasonal Salt Pan Crust Dynamics and Climatic Drivers Using Landsat 8 Time-Series and Ground Data. Remote Sens. 12, 474. https://doi.org/10.3390/rs12030474, 2020.

**Specific comments**

**Abstract**

Line 14    "The proposed methodology leverages on two complementary parts exploiting simulated and imagery data acquired over two study areas" What is the benefit in simulating Landsat-8 OLI data that are readily available globally? Furthermore, when an image close to the ground sampling date of the field sampling is available (according to Bannari & Al-Ali (2020))

Answer:
Dear reviewer, in this study and in remote sensing in general, the benefit and interest of spectral measurements simulating a specific satellite data is a strong and rigorous validation using as well the field truth sampling and the laboratory analysis. **Moreover, I would like to clarify that there is NO link between this research paper and the paper published by Bannari and Al-Ali (2020)**. In fact, even the SSSI model was used, the paper **Bannari and Al-Ali (2020)** was about climate change impact on soil salinity during 30 years using **ONLY Landsat image data acquired by three different sensors: TM, ETM+ and OLI**. The only intersection between these works is the considered study site.

Line 29    "Although the Omongwa [pan] is a natural flat salt playa, the four derived VI's from OLI image are completely unable to detect the slightest grain of salt in the soil" Why should vegetation indices that are parametrizations of the difference between

red/infrared bands and "greenness" detect salt that is mostly featureless in the VNIR? This consideration has no basis at all.

Answer:
Dear reviewer, as a remote sensing specialist we know well that it is impossible for VI's to map soil salinity. This was anticipated and this is the information that we want to show for people who use remote sensing as a tool. Because, as explain in our paper, in the literature many scientists claim the predictive power of VI's for soil salinity discrimination and mapping: **which a WRRONG statement**. For instance, Allbed et al. (2014) found that the SAVI extracted from the IKONOS image is useful for assessing the soil salinity in areas dominated by date palm trees. On the other hand, over agricultural lands in South Dakota Lobell et al. (2010) observed that EVI was significantly correlated with soil salinity and more sensitive to changes in salinity stress than NDVI. Moreover, Whitney et al. (2018) observed that the strength of the correlation coefficients between VIs and salinity was generally better for NDVI than for EVI. ETC……

**Introduction:**

Line 41        The first paragraph introduces the problem/ threat of salinization in well written from. However, no link to the study regions is provided. Is salinization a relevant problem for the areas under study? E.g., the Omongwa salt pan is a natural salt pan only with minor anthropogenic impact that is not used for any agriculture that could suffer from the stated problem of salinization.

Answer:
Dear reviewer, this comment is really very very bizarre !!!? because the introduction is very clear and it is not linked to specific study site. It provide relevant information about how soil salinity around the world is a serious problem and it is not relevant. In addition, at this section we are really so fare from the study site Omongwa salt-pan description or argumentation about the reasons behind its selection as a reference in our study.

Line 61-66    Please follow a consistent format for the ion's indication of electrical charge (either always with or without indication).

Done.

Line 75        This sentence needs rephrasing, as EC measurements are rather cheap compared to most laboratory soil analysis. Indeed, for larger areas/repeated measurements costs for sampling activities in relevant, but it is not the laboratory method that is expensive.

Answer:
Dear reviewer, I am so sorry, this phrase is correct 100%. Indeed, we are discussing about EC-$_{Lab}$ determined through the laboratory analysis using water extracted from a saturated soil paste which is globally accepted a standard to quantify soil salinity.

Line 77        "image processing methods have outperformed ground-based methods" is a very strong claim, as ground based, laboratory measurements they are definitely more accurate compared to remote sensing based retrievals. Suggestions: "more suitable" or "practical"

Answer:
Dear reviewer, I do not agree with your opinion because it is the reality and the truth for large area. Moreover, this is not only my opinion but also that of the scientific community, which is working on soil salinity using remote sensing technology. I invite you to read this book, it is so relevant regarding your comment:

Metternicht, G., and Zinck, J.A.: Remote Sensing of Soil Salinization: Impact on Land Management. CRC Press Taylor and Francis Group, Boca Raton, FL, USA, 374 pp., 2009.

Line 134     "acquired with diver sensors" Most likely "different sensors" is meant?

Done, thanks

Line 137     "Coincidentally, the NDGI is simply the SI-ASTER-4,5 proposed 16 years ago by Al-Khaier (2003)." SI-ASTER-4,5 and the NDGI seem very different. Whereas, the NDGI defined in Milewski et al. (2019) is a band ratio that exploits narrowband reflectances in the SWIR I (1.69 and 1.75 µm), which are both part of the 4th ASTER band (1.6-1.7µm). ASTER band 5 covers the parts of the SWIR II (2.145-2.185 µm) which is not part of the NDGI defined in Milewski et al. (2019).

Answer:
Dear reviewer, in remote sensing science and technology we analyze first the spectral signature looking for the absorption features characterizing the mineral under investigation as I explain above. Then, we select the spectral bands of interest depending on the used sensor. Obviously, the sensor bands are relatively different because they do not have the same width. For instance, the equation of NDVI still the NDVI independently to the sued satellite sensors (MODIS, MSS, TM, ETM+, OLI, SPOT, ASTER, Etc.....). The only condition is the inter-calibration between sensors if we are using many sensor. If we are using only one sensor data the NDVI still the NDVI. **Without any doubt the NDGI is simply the SI-ASTER-4,5 proposed 16 years ago by Al-Khaier (2003)**

Line 165     "Cert" typo? Is "Certainly" meant?

Done.

**Materials and Methods**

Line 202     Elaboration is needed on how the "CAM5S radiative transfer code" has been used in the data processing of the laboratory spectra. Normally, radiative transfer is used for atmospheric modelling to convert TOA radiance to surface reflectances (e.g., described later on starting at Line 340), but laboratory spectral measurements are used that are not affected by atmospheric disturbances. The convolution of the spectra to different spectral resolution is usually not linked to "radiative transfer" techniques, but simply done by applying the published relative spectral response factors (e.g., in 1 nm resolution) of the OLI sensor to the laboratory spectra.

Answer:
Dear reviewer, **this comment is WRRONG**. I am retired professor and my entire carrier during 35 years was in remote sensing domain. First, I want to tell you that the CAM5S (Teillet and Santer, 1991) is an improvement of 5S (Tanre et al., 1990) and relatively similar

to 6S (Vermote et al., 1997) that are the most used RTC by NASA not only for atmospheric modelling but also for **new sensor simulation and design**. As well as, the resampling operation **IS ALSO LINKED to the RTC based on the responsivity filters**. Please, read the following references:

- Tanre, D., Deroo, C, Duhaut, P., Herman, M., Morcrette, J.1., Perbos, J. and Deschamps, P.Y. (1990). *Simulation of the Satellite Signal in the Solar Spectrum (5S)*. Laboratoire d'Optique Atmospherique, Universite des Sciences et Techniques de Lille, Lille, France, 320 pages.
- Teillet, P.M., and Santer, R.: Terrain Elevation and Sensor Altitude Dependence in a Semi-Analytical Atmospheric Code". Canadian J. of Remote Sens. 17, 36-44, 1991.
- Vermote, E., Tanre, D., Deuze, J.L. et al. (1997) Second simulation of the satellite signal in the solar spectrum, 6S: An overview," IEEE Transactions on Geo-science and Remote Sensing, 1997, 35(3): 675-686.

Line 337      The SWIR bands of OLI do not cover the wavelength used in the NDGI proposed in Milewski et al. (2019), which exploits narrow spectral bands (10-20 nm) for gypsum detection, specifically the shoulder of the SWIR I absorption at 1690 nm and the center of the absorption feature at 1750 nm. Figure 5 shows very well that the sensitivity of the SWIR-1 Landsat band ends even before the shoulder of the absorption feature. The normalized SWIR I + SWIR II multispectral band ratio used here is commonly known under the name of Normalized Burn Ratio-2 (NBR2) as spectral indices routinely provided by the Landsat mission (https://www.usgs.gov/core-science-systems/nli/landsat/landsat-normalized-burn-ratio-2) and used for over two decades in the field of soil remote sensing (van Deventer et al. 1997).

van Deventer, A.P., Ward, A.D., Gowda, P.H., Lyon, J.G., 1997. Using Thematic Mapper Data to Identify Contrasting Soil Plains and Tillage Practices. American Society for Photogrammetry and Remote Sensing, Photogrammetric Engineering & Remote Sensing Vol. 63 No. 1 pp. 87-93.

Answer:
Dear reviewer, I answered for this point before. Please, to understand this process read the following paper where the same authors (Milewski et al., 2020) are using the same index (NDGI) and Landsat-OLI data:

Milewski, R., Chabrillat, S., and Bookhagen, B.: Analyses of Namibian Seasonal Salt Pan Crust Dynamics and Climatic Drivers Using Landsat 8 Time-Series and Ground Data. Remote Sens. 12, 474. https://doi.org/10.3390/rs12030474, 2020.

**Results Analysis**

Line 420      The mineralogy of sample "H" cannot be only "a pure salt-sabkha (bright florescent halite)", as pure halite would be featureless (except water absorption) and the shown spectrum certainly has strong absorption features. This spectrum could contain abundances of bischofite or other MgCl salts (compare spectra of https://10.1016/j.geoderma.2008.03.011 or doi.org/10.3390/rs6042647).

Answer:

Dear reviewer, why sample "H" can not be only a pure salt-sabkha ??????!!!!!!!. Always the halite crust is accumulated on the surface, while the other minerals (less soluble) are precipitated under the layer of halite. Moreover, according to the laboratory analysis preformed and used in this study, the validation zone surrounding sample point 141 is nearly pure halite (94%) mixed with very low content of gypsum (3%). In the first study site (Figure 2) the photo f is pure halite, as well in the second study site (Figure 3) the photo C is pure halite!!!!!!!

Line 503    "Consequently, undoubtedly VI's cannot exhibit the spatial patterns variability or provide precise and reliable information about the soil salinity. This finding corroborate those of spectral and CRRS analyses." This statement is only true for bare soil/sediment surfaces. Obviously, for none-vegetate areas case VI cannot be used to directly map top-soil salinity. But a much more reasonable use for VI is by proxy of salinity by degradation of soil fertility and vegetation vitality. E.g., Zhu et al. (2020, doi.org/10.3390/rs13020250) showed that spectra of agricultural crops do reflect the physiological changes of crops under soil salt stress, and it is feasible to model soil salinity using spectral vegetation indices of the crop canopy (e.g., NDVI with $R^2$ of 0.78 and RMSE (g/L) of 0.62 TDS). Similarly, a study of Al-Khakani & Yousif (2019, doi.org/10.1088/1742-6596/1234/1/012023) demonstrates an inverse correlation between the soil salinity and NDVI vegetation cover up to $R^2$ of 0.88 for a larger regional mapping. These examples show that VI can be largely beneficial for salinity mapping and that the usefulness depends on the surface type and specific application.

Answer:
Dear reviewer, the sentence from line 503 to 507 **is correct 100%** and it express the conclusions about this study regarding soil salinity discrimination and mapping and NOT soil degradation, fertility, or vegetation vitality.

Figure 7    Why are soils with 0.2-0.3 reflectance in the red band annotated with "bright crust" and soils with 0.6 reflectance as "dark soil"? Are the salt crusts not brighter throughout the VIS compared to none-saline soils?

Answer:
You are right; many of used samples are collected from sandy soil with high mixture and colored salt and not brighter (Please, see the Figure 2, photo e). This illustration is well known in the literature. Please, see the following reference:

Bannari, A., Huete, A. R., Morin, D., and Zagolski, F. (1996) Effects of soil color and brightness on vegetation indices.  Int. J. of Remote Sens., 17(10), 1885-1906.

Line 516    "evaporite minerals indices have the highest power for soil salinity discrimination with R2 517 of 0.71 and 0.72 for NDGI (or SI-ASTER-4,5) and SSSI (Figs. 8e and 8f), respectively. These results are due to the absorption features of salts (gypsum, halite, etc.) in SWIR bands, which are integrated in the equations of the both indices." In this statement, an important misunderstanding becomes apparent that leads to a misinterpretation of the actual salinity of the salt pan. The correlation between broadband SWIR indices and general soil salinity is highly dependent on the specific salt mineralogy and its effects on the spectrum. All surfaces that have higher SWIR II absorption than SWIR I would result in a high values of the SI-ASTER-4,5 and SSSI.

However, this is not the case for one of the most important/strongest cause of soil salinity at the test site: NaCL (halite), which is almost featureless in the optical range. This important limitation in the applied method needs to be addressed and the interpretation needs to be revised accordingly. For a representation of halite distribution at the Omongwa salt pan see the spectral unmixing result of Milewski et al. (2017).

Answer:
Dear reviewer, I am so sorry the analysis is clea,r well developed, and we do not any misunderstanding in this section. In addition, you must understand that using multispectral remote sensing it is not possible to discriminate among minerals. This statement is well known in the basic remote sensing books and courses. Contrary to hyperspectral remote sensing which provide the possibility and the power to discriminate among salt mineral particularly when using unmixing with endmember. Indeed, this was accomplished in the work that you are referring to: Milewski et al. (2017):

Milewski, R., Chabrillat, S., and Behling, R.: Analyses of Recent Sediment Surface Dynamic of a Namibian Kalahari Salt Pan Based on Multitemporal Landsat and Hyperspectral Hyperion Data. Remote Sens. 9, 170. https://doi.org/10.3390/rs9020170, 2017.

Line 588      "the SSSI further highlights the main salt crusts present in the pan area that are formed from different mineral sources, including halite, gypsum, calcite, and sepiolite" No, unfortunately the main areas of ~pure halite crust is missed by the SWIR indices. Both indices highlight the presents of gypsum that certainly increases general soil salinity, but not as much as halite. Most concerning, the halite rich areas are mapped as none saline! See comment above. A simple brightness/albedo based spectral index would lead to a more accurate representation of soil salinity, when halite is the major cause.

Answer:
Dear reviewer, at this step of analysis, we are analyzing the derived result using a semi-empirical model based on the SWIR bands BUT NOT only the SWIR as a single band. Moreover, or analysis is based obtained maps and validated based on the field truth and laboratory analysis (Table 2) which provide us the percentage of each salt-mineral.

Line 603      "a total absence of salinity is observed in the center and center-east parts of the pan (black pixels) due to the presence of water which absorbs the signal in the SWIR wavelengths." No, this image acquired at the dry season does not have any surface water present. Please check the Landsat spectra or simply a RGB representation. These areas that appear low in SWIRI-SWIRII indices are the most saline pixels that have a much brighter, but flat spectrum in the SWIR. See attached image of the Landsat OLI scene and exemplary spectra.

Answer:
Dear reviewer, **I am so sorry to say that your statement is WRRONG**. If you know remote sensing science, you must understand the basic concepts. Indeed, the soil moisture and shallow water surfaces are absorbed in the SWIR wavelengths and not in the visible (RGB) bands. Second, if you read carefully the paper you will see that presented RGB image for study site illustration is not the one used in this research analysis. Third, the dry season does not mean the absence of water accumulated in low areas of the study site as illustrated by the Lidar DEM (Figure 9).

**Discussion**

Line 666    Here, the fact that halite is featureless is mentioned, but is not recognized as a limitation for the applied methods and the results are not discussed according.

Answer:
This sentence is related to the one before; it is clear and it explain the absence of pure halite features in the VNIR.

Line 676    "While when the EC-Lab values increased also the difference among the salt-affected soil spectra's increased significantly and progressively in the SWIR"

Done.

Line 691    "SWIR-1 and SWIR-2 bands of OLI show the highest potential to discriminate efficiently among different degrees of salinity in the soil." These statements are valid for some salt minerals, e.g., gypsum or bischofite, but not true at all for other salt minerals, e.g., halite or thenardite. Consequently, SWIR1-SWIR2 indices might produce totally misleading salinity maps for test sites that are for example dominated by halite.

Answer:
Dear reviewer, I agree with you and this the reason why we use the spectral measurements for absorption features. However, the equations of the both indices are integrating the SWIR bands but the difference in the conception of their models allows highlighting the difference between gypsum and/or other salts minerals.

Line 683    As stated in the comment above sample "H" has several features that are not related to halite and spectrally indicate the prancer of a different salt mineral.

Answer:
Dear reviewer, in the line 683 we say: "For instance, CRRS analyses of halite-rich soil "H" sample showed consistent absorption features at 960, 1160, 1420, 1780 and 1920 nm and they become deeper, broader, and more asymmetrical with increasing salt content in the soil". This statement is correct because we have a dominant halite class mixed with other non-dominant classes. For instance, Table 2 for the soil laboratory analyses shows you the ponderation of classes from a sampling point to another.

Line 693    "of LAI at different densities confirmed the relevance of [..] stress." This dataset of LAI measurements might be undervalued in the current pre-print status. Instead of calculating vegetation indices for bare soil, the LAI or vegetation indices of the Kuwait imagery (not lab spectra) might show a correlation to the soil salinity measurements, due to reduction in vegetation vitality.

Answer:
Dear reviewer, I am so sorry your suggestion is not the idea of this paper.

Line 712    Here the higher albedo of the central pan area is actually mentioned, but no contradiction with the results of the SWIR salinity indices is noted or discussed.

Answer:

Done, the SWIR idea is linked in the section between lines 728-746.

Suggested to revise some dramatic wording to more objective language, e.g., Line 559: "Faithful to their mission" or Line 555: "indices are blind"

Answer:
Dear reviewer, these words are correct linguistically, they describe correctly and perfectly the ideas that we want to convey in this research.

**Dear reviewer many thanks for your time and your comments.**

Professor A. Bannari

https://www.researchgate.net/profile/Bannari-Abdou

https://scholar.google.fr/citations?user=D6sZhSQAAAAJ&hl=fr

---

## Author Comment (AC3)

This publication addresses a global environmental issue that affects a large portion of soils around the world. Salinity as a natural and anthropic phenomenon has been widely studied by geologists and in particular by geochemists, but the evaluation of its extent and the monitoring of its dynamics have not known the same scientific interest.

Another contribution of this article is the understanding of the dynamics of salts in soils; its strong contribution is the proposal and validation of a method of mapping and monitoring soil salinity based on new indices other than those usually known. I think that the choice of two saline environments with different geochemical dynamics was an excellent idea to test the limits of the proposed method.

**Dear reviewer thank you for time and your positive words.**

Professor A. Bannari

https://www.researchgate.net/profile/Bannari-Abdou

https://scholar.google.fr/citations?user=D6sZhSQAAAAJ&hl=fr